# Trifuse: Enhancing Attention-Based GUI Grounding via Multimodal Fusion

Longhui Ma [1] [*]   Di Zhao [1] [*]   Siwei Wang [2]   Zhao Lv [2]   Miao Wang [2]

## Abstract

GUI grounding maps natural language instructions to the correct interface elements, serving as the perception foundation for GUI agents. Existing approaches predominantly rely on fine-tuning multimodal large language models (MLLMs) using large-scale GUI datasets to predict target element coordinates, which is data-intensive and generalizes poorly to unseen interfaces. Recent attention-based alternatives exploit localization signals in MLLMs attention mechanisms without task-specific fine-tuning, but suffer from low reliability due to the lack of explicit and complementary spatial anchors in GUI images. To address this limitation, we propose Trifuse, an attention-based grounding framework that explicitly integrates complementary spatial anchors. Trifuse integrates attention, OCR-derived textual cues, and icon-level caption semantics via a Consensus-SinglePeak (CS) fusion strategy that enforces cross-modal agreement while retaining sharp localization peaks. Extensive evaluations on four grounding benchmarks demonstrate that Trifuse achieves strong performance without task-specific fine-tuning, substantially reducing the reliance on GUI-specific annotated grounding data. Moreover, ablation studies reveal that incorporating OCR and caption cues consistently improves attention-based grounding performance across different backbones, highlighting its effectiveness as a general strategy for GUI grounding.

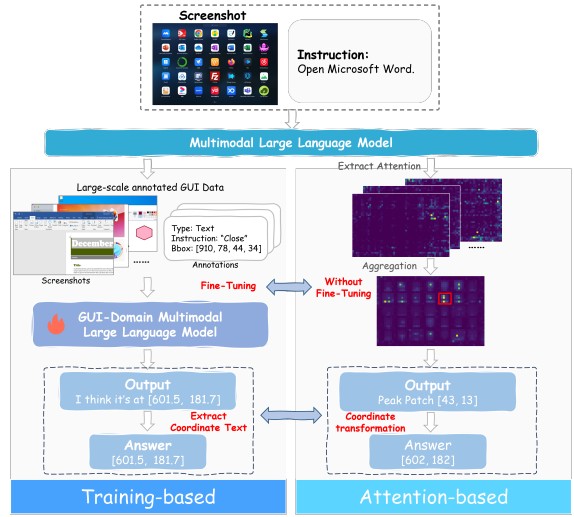

*Figure 1.* Comparison between training-based and attention-based methods for GUI grounding.

## 1. Introduction

Graphical User Interfaces (GUIs) represent an essential component of contemporary digital systems, enabling users to interact visually with software through GUI elements (Shneiderman, 2010; Johnson, 2020). Recent advances in multimodal large language models (MLLMs) (Hurst et al., 2024; Team et al., 2023; Bai et al., 2025) have driven increased research interest toward the development of intelligent GUI agents capable of understanding and interacting within GUI environments (Wang et al., 2024c; Zhang et al., 2024). These agents offer significant potential for automating software tasks, supporting accessibility requirements, and enabling generalization across diverse platforms, thereby opening new avenues for advancements in human-computer interaction (Lin et al., 2024; Gou et al.).

Accurately grounding natural language instructions to corresponding GUI elements remains a critical requirement for GUI agents (Tang et al., 2025b). Recent work predominantly employs training-based methods, where MLLMs are fine-tuned on large-scale GUI datasets to directly predict target element coordinates. While these methods (Lin et al., 2024; Qin et al., 2025) achieve strong grounding benchmark performance (Cheng et al., 2024; Xie et al., 2025), they require substantial annotated data and often exhibit

*Equal contribution [1] College of Computer Science and Technology, National University of Defense Technology, Changsha, Hunan, China [2] Intelligent Game and Decision Lab, Academy of Military Sciences, Beijing, China . Correspondence to: Siwei Wang <wangsiwei13@nudt.edu.cn>.

*Proceedings of the 43rd International Conference on Machine Learning*, Seoul, South Korea. PMLR 306, 2026. Copyright 2026 by the author(s).

limited generalization. Adapting to new applications, layouts, or operating systems typically necessitates additional data collection and retraining (Lu et al., 2025a). Attention-based methods offer an alternative by leveraging the internal attention mechanisms of MLLMs for localization. Prior work demonstrates that MLLMs' attention naturally focuses on task-relevant visual regions in natural images (Zhang et al.; Kang et al., 2025). TAG (Xu et al., 2025) first applies this observation to GUI environments by selecting informative tokens and attention heads to construct attention maps for grounding without task-specific fine-tuning. While demonstrating the feasibility of attention-based grounding, TAG still lags behind supervised models in grounding performance. Subsequent attention-based methods, including GUI-Actor and GUI-AIMA (Wu et al., 2025b; Zhou et al., 2025a), introduce learnable tokens to guide attention, improving accuracy but reintroducing training requirements and data dependencies.

This naturally raises a key question: *Can accurate GUI grounding be achieved using attention mechanisms without task-specific fine-tuning?* We believe that the failure of attention-based GUI grounding is not due to a lack of semantic understanding, as TAG (Xu et al., 2025) has already shown promising grounding capability, but rather to a lack of explicit and complementary spatial anchors. Recent studies (Lu et al., 2024; Wang et al., 2025; Wu et al., 2025a) show that incorporating additional GUI information—such as text, visual semantics, and structural relations—substantially improves grounding performance, motivating a multimodal fusion approach. Although MLLM attention implicitly encodes textual and semantic information, these signals tend to be spatially diffuse and entangled across attention heads. In contrast, explicit OCR and caption modalities provide localized semantic anchors that are easier to align with GUI regions. Building on these insights, we propose Trifuse, an attention-based GUI grounding framework that integrates three complementary modalities: MLLM attention, OCR-based textual cues and icon caption-based visual semantics. For each modality, Trifuse uses dedicated mechanisms to extract corresponding heatmaps. Then, Trifuse fuses these modality heatmaps through a Consensus-SinglePeak (CS) strategy that emphasizes consensus signals across modalities while retaining sharp localization peaks. Compared with other attention-based methods, experiments show that Trifuse outperforms prior methods (Xu et al., 2025) on four grounding benchmarks (Cheng et al., 2024; Li et al., 2025; Xie et al., 2025). Furthermore, when integrated with training-based models (Wu et al., 2025b; Zhou et al., 2025a), Trifuse further improves localization accuracy, demonstrating the complementary and generalizable value of multimodal fusion even for fine-tuned MLLMs. Our contributions are:

- We propose Trifuse, a simple yet effective multimodal fusion framework for attention-based GUI grounding that fuses three modalities: MLLM attention, OCR-based textual cues and icon caption-based visual semantics, improving grounding accuracy without requiring task-specific fine-tuning.

- We introduce an effective attention extraction strategy that selects informative tokens and attention heads from MLLMs, enabling more precise and robust grounding.

- We design a Consensus-SinglePeak (CS) fusion strategy that jointly models cross-modal agreement and modality-specific discriminative signals, effectively suppressing noise while preserving informative peaks. Extensive experiments across multiple benchmarks validate the effectiveness and generality of the proposed fusion strategy.

**Conflict of Interest Disclosure.** The authors declare no financial conflicts of interest.

## 2. Related Work

### 2.1. GUI Agents

Recent advances in large language models (LLMs) and vision-language integration have enabled powerful MLLMs such as GPT-4o (Hurst et al., 2024), Gemini (Team et al., 2023), and open-source alternatives like Qwen-VL (Wang et al., 2024b; Bai et al., 2025), which demonstrate strong capabilities across vision-language tasks. These MLLMs provide a foundation for GUI agents that understand user instructions and interact with GUIs to complete tasks. Early GUI agents primarily operated in web and mobile platforms with access to structured representations such as HTML DOM trees or accessibility hierarchies (Deng et al., 2023; Wang et al., 2024a), enabling precise element identification through explicit structural information. Recent work has shifted toward vision-centric paradigms that rely solely on visual observations (Lin et al., 2024; Gou et al.; Xu et al., 2024), eliminating dependence on platform-specific APIs or structured data.

### 2.2. Grounding in GUI Agents

GUI grounding aims to map natural language instructions to target GUI elements in screenshots. Training-based methods are the most commonly used approaches, which fine-tune MLLMs on large GUI datasets to predict element coordinates (Xu et al., 2024; Cheng et al., 2024; Wu et al.). While achieving strong benchmark performance (Qin et al., 2025), these methods require substantial annotated data to align spatial coordinates with visual elements and often generalize poorly without additional retraining. Reinforcement learning approaches (Li & Huang, 2025) attempt to reduce data requirements through GUI-specific reward signals

but remain computationally intensive. Training-free methods attempt to locate target elements without task-specific fine-tuning (Lu et al., 2024; Wu et al., 2025a; Yang et al., 2023). Among these methods, attention-based methods exploit the internal attention mechanisms of MLLMs, which have been shown to naturally focus on task-relevant visual regions (Zhang et al.; Kang et al., 2025). TAG (Xu et al., 2025) first exploits this observation in GUI environments, demonstrating that MLLMs' attention possesses inherent GUI grounding capability. However, performance is limited by heuristic token and head selection strategies. To improve accuracy, subsequent attention-based methods introduce specialized learnable tokens (Wu et al., 2025b; Zhou et al., 2025a), but require task-specific fine-tuning.

# 3. Method

Trifuse mainly consists of three parts: modality extraction, modality fusion, and localization. The overall framework is illustrated in Figure 2. The modality extraction stage constructs three complementary grounding heatmaps: (1) attention-based heatmap via fine-grained token and head filtering, (2) OCR-based heatmap from detected textual elements, and (3) caption-based heatmap from icon-level visual semantics. The modality fusion stage integrates these modalities through the Consensus-SinglePeak (CS) strategy. Finally, a two-stage localization strategy is applied to the fused heatmap to locate the target element.

## 3.1. Modality Extraction

### 3.1.1. ATTENTION MODALITY

MLLMs produce multi-head attention maps across layers. However, directly leveraging all attention information for grounding performs poorly due to noise from irrelevant tokens and weakly-aligned heads (Xu et al., 2025). We address this through a two-level filtering scheme: token-level filtering identifies task-relevant query tokens, and head-level filtering identifies spatially informative attention heads. The filtered attention signals are subsequently aggregated to construct a reliable attention-based grounding heatmap.

**Preliminaries** We consider an MLLM with $L$ layers and $H$ attention heads that processes GUI screenshots $V$ and user instructions $Q$. $V$ is represented as a sequence of visual patch tokens $\mathcal{V} = [v_1, \ldots, v_{|\mathcal{V}|}]$ after the vision encoder. The user instructions are tokenized as $\mathcal{Q} = [q_1, \ldots, q_{|\mathcal{Q}|}]$. At layer $l$ and head $h$, the model computes a self-attention matrix $\mathbf{A}^{l,h}$. For each query token $q_i$, we extract its patch-wise attention vector $\mathbf{a}_{q_i}^{l,h} \in \mathbb{R}^{|\mathcal{V}|}$, where the $j$-th element of $\mathbf{a}_{q_i}^{l,h}$ indicates how strongly $q_i$ attends to visual patch $v_j$.

**Token-level Filtering** User instructions typically consist of multiple tokens with varying relevance to the target ele-

ments. While certain tokens explicitly describe the target object or its attributes, others are only implicitly related, and function words such as prepositions or articles often provide little grounding signal. Prior studies on referring expression comprehension and vision–language grounding have shown that treating all tokens equally introduces substantial noise and degrades localization performance (Yu et al., 2018; Qiao et al., 2020). This motivates the need for selectively emphasizing informative tokens during grounding.

To identify the most relevant tokens, we measure the alignment between each text token $q_i$ and the visual content by computing a token-image relevance score:

$$S(q_i) = \sum_{v_j \in \mathcal{V}} \mathbb{I}[\cos(q_i, v_j) > \tau_v] \cdot \cos(q_i, v_j), \quad (1)$$

where $q_i$ and $v_j$ denote the embedding of token $q_i$ and visual patch $v_j$, respectively. A higher score indicates stronger correlation with image regions, suggesting greater relevance for grounding. $\tau_v$ reduces interference from irrelevant patches. Then, we retain the top-$k$ tokens with the highest relevance scores, forming the filtered token set $\mathcal{Q}_{\text{filter}}$.

**Head-level Filtering** Recent work has shown that not all attention heads are equally suitable for visual grounding in natural images (Xiao et al., 2024; Kang et al., 2025), and similar observations hold in the GUI domain (Xu et al., 2025). Only a subset of heads exhibits spatially concentrated attention patterns useful for grounding, while others distribute attention diffusely across the image (see Appendix E.2 for cases).

To identify grounding-relevant attention heads, we use spatial entropy to quantify how concentrated each head's attention distribution is over spatial regions. Specifically, we group the attention heatmap into a set of spatially connected regions using connected component analysis implemented via a union–find algorithm, denoted as $\mathcal{R} = \{r_1, r_2, \ldots, r_{|\mathcal{R}|}\}$. For a given attention head $(l, h)$ and a selected token $q_i \in \mathcal{Q}_{\text{filter}}$, the spatial entropy of its attention distribution $\mathbf{a}_{q_i}^{l,h}$ is defined as:

$$H\big(\mathbf{a}_{q_i}^{l,h}\big) = -\sum_{r_j \in \mathcal{R}} p(r_j) \log p(r_j) \quad (2)$$

where $p(r_j)$ denotes the normalized attention mass assigned to region $r_j$:

$$p(r_j) = \frac{\sum_{v_k \in r_j} \big(\mathbf{a}_{q_i}^{l,h}\big)_k}{\sum_{r_m \in \mathcal{R}} \sum_{v_k \in r_m} \big(\mathbf{a}_{q_i}^{l,h}\big)_k} \quad (3)$$

Lower entropy indicates that the attention is concentrated on fewer connected regions, suggesting that the head is more likely to focus on the target GUI elements. We retain the top-$k$ attention heads with the lowest spatial entropy, forming

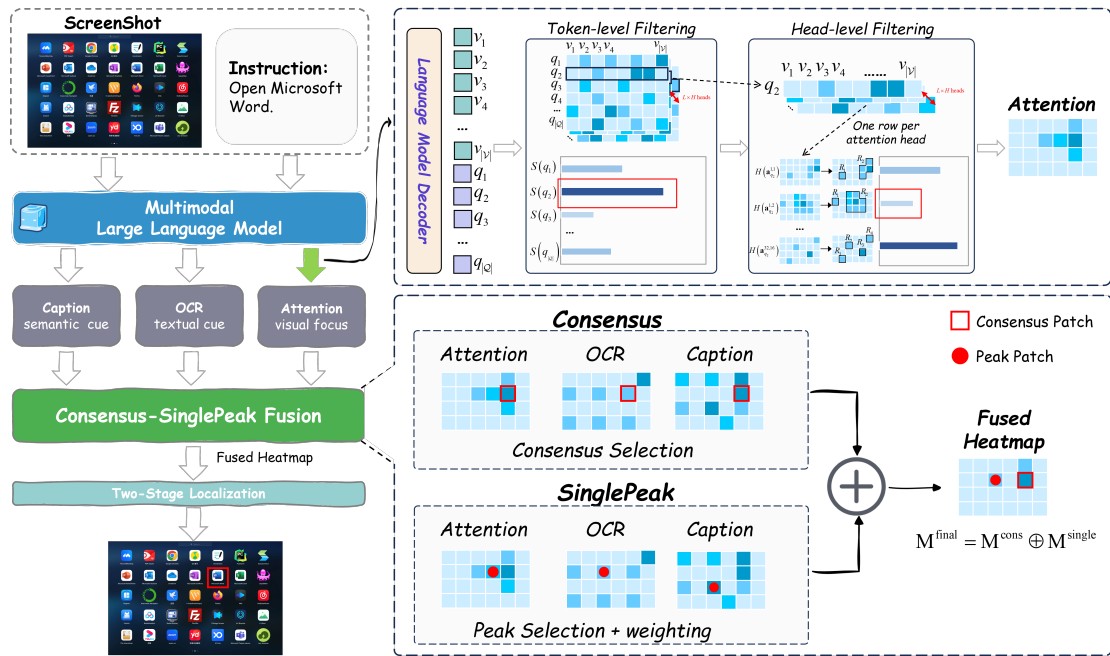

*Figure 2.* Overview of our Trifuse framework. Trifuse consists of three main components: (1) a modality extraction module that derives complementary grounding cues, including attention-based signals from MLLMs, textual cues from OCR, and icon-level visual semantics from captioning; (2) a Consensus-SinglePeak (CS) fusion module that integrates these modality-specific heatmaps by jointly modeling cross-modal agreement and modality-specific discriminative peaks; and (3) a two-stage localization module that progressively refines the fused grounding map through cropping and zoom-in operations to accurately identify the target GUI element.

the grounding-relevant head set $\mathcal{H}_{\text{filter}}$. Head selection is performed independently for each token in $\mathcal{Q}_{\text{filter}}$.

**Weighted Aggregation**     After token and head filtering, we construct the final attention heatmap $\mathbf{a}^{\text{attn}}$ via weighted aggregation over $\mathcal{Q}_{\text{filter}}$ and $\mathcal{H}_{\text{filter}}$. Specifically, token weights are computed by applying a softmax over token-image relevance scores, while head weights are obtained by applying a softmax over the negative spatial entropy.

$$\mathbf{a}^{\text{attn}} = \sum_{q_i \in \mathcal{Q}_{\text{filter}}} \text{softmax}(S(q_i)) \sum_{(l,h) \in \mathcal{H}_{\text{filter}}(q_i)} \text{softmax}\big(-H(\mathbf{a}_{q_i}^{l,h})\big) \mathbf{a}_{q_i}^{l,h} \quad (4)$$

### 3.1.2. OCR AND CAPTION MODALITY

**OCR Modality.**   GUI images often contain rich textual information, which plays a critical role in text-related grounding tasks (Singh et al., 2019). Accurately extracting and aligning textual content from GUI screenshots is therefore essential for precise grounding.

We employ an off-the-shelf OCR engine to extract textual elements from a given GUI image $V$. The OCR engine outputs a set of detected text instances $\mathcal{T} = \{t_1, t_2, \ldots, t_{|\mathcal{T}|}\}$, where each instance $t_k = \{b_k, s_k, c_k\}$ consists of a bounding box $b_k$, a recognized text string $s_k$, and a detection confidence score $c_k \in [0, 1]$. To align OCR results with the user instructions, we compute the semantic relevance

between each recognized text string and the filtered query tokens. Specifically, the OCR relevance score for instance $t_k$ is defined as:

$$r_k^{\text{ocr}} = \cos\big(\mathbf{e}(s_k), \mathcal{Q}_{\text{filter}}\big) \cdot c_k, \quad (5)$$

where $\mathbf{e}(\cdot)$ denotes the text embedding function. We then project the OCR relevance scores onto the visual patch grid. For each visual patch $v_j$, the OCR-based heatmap value aggregates contributions from all OCR bounding boxes that overlap with $v_j$:

$$a_j^{\text{ocr}} = \sum_{k:\, v_j \cap b_k \neq \emptyset} r_k^{\text{ocr}}. \quad (6)$$

After normalization, the resulting vector $\mathbf{a}^{\text{ocr}} \in \mathbb{R}^{|\mathcal{V}|}$ represents the OCR-based heatmap.

**Caption Modality.**   GUI screenshots contain abundant non-textual visual elements, such as buttons and graphical controls, which are essential for grounding instructions that do not explicitly mention text (Rawles et al., 2023; Deka et al., 2017). To capture such visual semantics beyond textual cues, we employ a pretrained icon detection model to localize GUI elements and generate semantic descriptions.

Given an input image $V$, the icon detection model outputs a set of detected icon instances $\mathcal{C} = \{c_1, c_2, \ldots, c_{|\mathcal{C}|}\}$, where each instance $c_k = \{b_k, d_k, c_k\}$ consists of a bounding box

$b_k$, a generated caption or semantic description $d_k$, and a detection confidence score $c_k \in [0, 1]$. To align icon semantics with the user instructions, we compute the semantic similarity between each generated caption and the filtered query tokens. The caption relevance score for instance $c_k$ is defined as:

$$r_k^{\text{cap}} = \cos\big(\mathbf{e}(d_k), \mathcal{Q}_{\text{filter}}\big) \cdot c_k. \tag{7}$$

Similar to the OCR modality, we project the caption relevance scores onto the visual patch grid. For each visual patch $v_j$, the caption-based heatmap value aggregates contributions from all detected icon regions that overlap with $v_j$:

$$a_j^{\text{cap}} = \sum_{k:\, v_j \cap b_k \neq \emptyset} r_k^{\text{cap}}. \tag{8}$$

After normalization, the resulting vector $\mathbf{a}^{\text{cap}} \in \mathbb{R}^{|\mathcal{V}|}$ represents the patch-wise caption grounding heatmap.

## 3.2. Modality Fusion

After obtaining the modality-specific heatmaps $\mathbf{a}^{\text{attn}}$, $\mathbf{a}^{\text{ocr}}$, and $\mathbf{a}^{\text{cap}}$, an effective fusion strategy is required to highlight the target region while avoiding bias toward any single modality. Simple averaging amplifies noise from dominant modalities, whereas consensus-based fusion risks discarding modality-specific cues. These limitations motivate a fusion strategy that explicitly separates cross-modal agreement from discriminative responses. Therefore, we propose a Consensus-SinglePeak (CS) fusion strategy that integrates two complementary types of signals: cross-modal consensus, which emphasizes regions consistently supported by multiple modalities, and modality-specific peaks, which preserve strong and discriminative responses unique to individual modalities. By jointly modeling these two aspects, CS fusion achieves a balanced trade-off between robustness and sensitivity.

Formally, the final fused heatmap is computed as:

$$\mathbf{M}^{\text{final}} = \mathbf{M}^{\text{cons}} \oplus \mathbf{M}^{\text{single}}, \tag{9}$$

where $\mathbf{M}^{\text{cons}} \in \mathbb{R}^{|\mathcal{V}|}$ captures the cross-modal consensus and $\mathbf{M}^{\text{single}} \in \mathbb{R}^{|\mathcal{V}|}$ preserves modality-specific peaks. The operator $\oplus$ denotes an element-wise fusion operation that combines the two signals by element-wise averaging.

**Consensus**  The consensus term identifies spatial regions where multiple modalities exhibit strong agreement. We compute the consensus heatmap by aggregating modality-specific heatmaps through element-wise multiplication:

$$M_j^{\text{cons}} = a_j^{\text{attn}} \odot a_j^{\text{ocr}} \odot a_j^{\text{cap}}, \tag{10}$$

where $a_j^{\text{attn}}$, $a_j^{\text{ocr}}$, and $a_j^{\text{cap}}$ denote heatmap values from attention, OCR, and caption modalities at patch position $j$, respectively. This multiplicative formulation amplifies regions that are consistently supported by all modalities.

**Single-Peak**  While the consensus term emphasizes regions jointly supported by multiple modalities, modality-specific peaks often provide crucial discriminative cues that may be absent from other modalities. For example, OCR can uniquely localize a target text element that is weakly attended by the attention or caption modalities. To preserve such informative signals while suppressing spurious responses, we introduce a confidence-aware single-peak selection mechanism.

For each modality $s \in \{\text{attn}, \text{ocr}, \text{cap}\}$ and patch position $j$, we first identify candidate peaks whose heatmap responses exceed a modality-specific threshold, i.e., $a_j^s > \tau_s$. Let $P_s = \{j \mid a_j^s > \tau_s\}$ denote the peak set for modality $s$. To estimate the reliability of each peak, we measure its relative support from the remaining modalities:

$$\text{conf}_{s,j} = \sigma\left(\alpha \cdot \frac{\sum_{s' \neq s} a_j^{s'}}{a_j^s + \varepsilon} - \beta\right), \tag{11}$$

where $\sigma(\cdot)$ denotes the sigmoid function, $\varepsilon$ is a small constant for numerical stability, and $\alpha, \beta$ control the sensitivity of the confidence estimation. Intuitively, the confidence score is high when a modality-specific peak is supported by non-negligible responses from other modalities, and low when the peak is isolated and potentially noisy.

Based on the confidence score, we assign a dynamic weight to each peak:

$$W_{s,j} = 1 + \lambda\big(2\,\text{conf}_{s,j} - 1\big), \tag{12}$$

where $\lambda \geq 0$ controls the strength of peak amplification. High-confidence peaks are amplified ($W_{s,j} > 1$), whereas low-confidence peaks are softly down-weighted ($W_{s,j} < 1$). The single-peak heatmap aggregates all weighted modality-specific peaks:

$$M_j^{\text{single}} = \sum_{s \in \{\text{attn}, \text{ocr}, \text{cap}\}} \mathbb{I}(j \in P_s)\, W_{s,j}\, a_j^s. \tag{13}$$

Together with the consensus term, the Single-Peak component enables CS fusion to retain highly discriminative modality-specific cues while maintaining robustness to noise. After obtaining the fused heatmap $\mathbf{M}^{\text{final}}$, we next describe how to localize the target GUI elements from it.

## 3.3. Two-Stage Localization

Directly selecting the peak from the fused heatmap $\mathbf{M}^{\text{final}}$ often yields suboptimal localization accuracy (See Appendix B.4), primarily due to the limited spatial granularity of visual tokens in MLLMs. High-resolution GUI screenshots are typically downsampled to meet GPU memory constraints, resulting in coarse patch tokenization that degrades fine-grained grounding.

To address this issue, we adopt a simple yet effective two-stage zoom-in localization strategy (Li et al., 2025; Zhang et al., 2025) without requiring additional training. In the first stage, we perform inference on the downsampled full screenshot to obtain the fused heatmap $M^{final}$, and identify a coarse target region $R_{coarse}$ by selecting the patch with the maximum response. In the second stage, we crop a window centered at $R_{coarse}$ from the original high-resolution image, resize it to the model's input resolution, and re-run inference to obtain a refined prediction. This hierarchical strategy decouples coarse region identification from fine-grained localization, substantially improving precision on high-resolution displays without architectural modifications or task-specific retraining.

## 4. Experiments

**Implementation Details.** We employ Qwen2.5-VL-3B-Instruct and Qwen2.5-VL-7B-Instruct (Bai et al., 2025) as the MLLM backbone for attention extraction. The model consists of 32 transformer layers, each with 16 attention heads. Following the token and head filtering strategy described in Section 3, we select the top-1 semantically relevant token and aggregate attention from the top-6 spatially informative heads. For the OCR and caption modality, we adopt PaddleOCR v4 (Cui et al., 2025) for text detection and recognition, and OmniParser (Lu et al., 2024) for icon localization and semantic caption generation, respectively. To compute semantic similarity between textual content (OCR results or icon captions) and the user instructions, we use the BGE-M3 (Chen et al., 2024) embedding model. For two-stage localization, we perform a single zoom-in iteration. Specifically, a cropped window of size $\frac{W}{2} \times \frac{H}{2}$ is extracted from the original image and centered at the coarse prediction, where $W \times H$ denotes the input image resolution. The cropped region is then resized to the model's input resolution for refined inference. We follow standard practice and define training-free as no task-specific fine-tuning on GUI grounding datasets. All auxiliary models (OCR, icon caption model, embedding models) are used off-the-shelf without adaptation.

**Evaluation Benchmarks** We evaluate Trifuse on four established GUI grounding benchmarks: ScreenSpot, ScreenSpot-v2 (Cheng et al., 2024), ScreenSpot-Pro (Li et al., 2025), and OSWorld-G (Xie et al., 2025). These benchmarks collectively assess grounding performance across diverse platforms (mobile, desktop, web), resolutions, and task complexities. We report Element Accuracy as the primary metric, which measures the proportion of predictions where the predicted click point falls within the ground-truth bounding box of the target elements. Additional benchmark details are provided in Appendix A.

*Table 1.* Performance comparison on four grounding benchmarks. TF denotes training-free methods, while TB denotes training-based methods. Δ denotes the improvement of Trifuse over the corresponding baseline.

| | Method | Model Size | ScreenSpot Avg. | Δ | ScreenSpot-v2 Avg. | Δ | ScreenSpot-Pro Avg. | Δ | OSWorld-G Avg. | Δ |
|---|---|---|---|---|---|---|---|---|---|---|
| TF | TAG (Xu et al., 2025) | 8.5B | 57.5 | - | 51.2 | - | 3.0 | - | 25.3 | - |
| | Trifuse | 3B | 81.1 | +23.6 | 82.6 | +31.4 | 18.9 | +15.9 | 38.7 | +13.4 |
| | Trifuse | 7B | 86.2 | +28.7 | 86.9 | +35.7 | 29.7 | +26.7 | 43.6 | +18.3 |
| TB | GUI-Actor (Wu et al., 2025b) | 3B | 86.5 | - | 91.0 | - | 42.2 | - | 54.6 | - |
| | Trifuse+GUI-Actor | 3B | 89.5 | +3.0 | 91.5 | +0.5 | 42.7 | +0.5 | 56.5 | +1.9 |
| | GUI-Actor (Wu et al., 2025b) | 7B | 88.3 | - | 92.1 | - | 44.6 | - | 56.6 | - |
| | Trifuse+GUI-Actor | 7B | 90.5 | +2.2 | 93.2 | +1.1 | 45.8 | +1.2 | 57.9 | +1.3 |
| | GUI-AIMA (Zhou et al., 2025a) | 3B | 88.1 | - | 91.5 | - | 49.8 | - | 58.3 | - |
| | Trifuse+GUI-AIMA | 7B | 90.6 | +2.5 | 92.9 | +1.4 | 51.3 | +1.5 | 58.4 | +0.1 |

**Baselines** We compare Trifuse against four categories of GUI grounding methods representing distinct paradigms. **General models** are pretrained vision-language models without GUI-specific fine-tuning, including GPT-4o (Hurst et al., 2024) and Qwen2.5-VL (Bai et al., 2025). **Supervised fine-tuning (SFT) models** are VLMs fine-tuned on GUI-annotated datasets, including UGround (Gou et al.), Aguvis (Xu et al., 2024), ShowUI (Lin et al., 2024), UI-TARS (Qin et al., 2025), and JEDI (Xie et al., 2025). **Reinforcement learning (RL) models** employ policy gradient methods with GUI-specific reward signals, including UI-R1 (Lu et al., 2025b), GUI-G1 (Zhou et al., 2025b), and GUI-G² (Tang et al., 2025a). **Attention-based methods** exploit attention mechanisms of MLLMs to localize target GUI elements. This category includes the training-free method TAG (Xu et al., 2025), as well as training-based approaches including GUI-Actor (Wu et al., 2025b) and GUI-AIMA (Zhou et al., 2025a).

Among these baselines, attention-based methods constitute our primary comparison group, as Trifuse similarly leverages attention for grounding while introducing multimodal fusion to enhance localization accuracy. To isolate the contribution of the proposed fusion strategy from backbone capacity, we evaluate Trifuse with multiple backbone MLLMs. Specifically, for comparisons with training-based methods, we replace the Qwen2.5-VL-3B-Instruct backbone with the corresponding fine-tuned GUI-Actor-3B, GUI-Actor-7B or GUI-AIMA-3B models, respectively. All other components of Trifuse, including token and head filtering, CS fusion, and two-stage localization, remain unchanged.

### 4.1. Main Results

We present the evaluation results of Trifuse against state-of-the-art GUI grounding methods across four benchmarks in Table 2, Table 3, Table 4 and Table 5. To facilitate a clearer comparison with attention-based approaches, Table 1 further summarizes the results under matched model backbones and evaluation settings. We emphasize pairwise comparisons under identical model settings and training paradigms, highlighting the consistent improvements brought by Trifuse over its attention-based counterparts rather than reporting

*Table 2.* Performance comparison on ScreenSpot. Training Data Size refer to the number of images used for training (expect UI-TARS).

| | Method | Training Data Size | Model Size | Mobile | | Desktop | | Web | | Avg. | Δ |
|---|---|---|---|---|---|---|---|---|---|---|---|
| | | | | Text | Icon | Text | Icon | Text | Icon | | |
| General | GPT-4o | - | / | 30.5 | 23.2 | 20.6 | 19.4 | 11.1 | 7.8 | 18.8 | - |
| | Qwen2.5-VL | - | 3B | 62.1 | 46.4 | 54.1 | 30.0 | 31.2 | 48.3 | 46.9 | - |
| | Qwen2.5-VL | - | 7B | 93.4 | 76.4 | 87.6 | 57.9 | 82.2 | 63.1 | 78.6 | - |
| | OmniParser | - | / | 93.9 | 57.0 | 91.3 | 63.6 | 81.3 | 51.0 | 73.0 | - |
| SFT | MP-GUI | 0.68M | 8B | 86.8 | 65.9 | 70.8 | 56.4 | 58.3 | 46.6 | 64.1 | - |
| | UGround | 1.3M | 7B | 82.8 | 60.3 | 82.5 | 63.6 | 80.4 | 70.4 | 73.3 | - |
| | ShowUI | 0.26M | 2B | 91.6 | 69.0 | 81.8 | 59.0 | 83.0 | 65.5 | 74.9 | - |
| | Aguvius | 1M | 7B | 78.2 | 88.3 | 70.7 | 88.1 | 74.8 | 85.7 | 81.8 | - |
| | UI-TARS | 50B Token | 2B | 93.0 | 75.5 | 90.7 | 68.6 | 84.3 | 74.8 | 82.3 | - |
| | UI-TARS | 50B Token | 7B | 94.5 | 85.2 | 95.9 | 85.7 | 90.0 | 83.5 | 89.5 | - |
| | JEDI | 1.39M | 3B | 96.9 | 81.5 | 96.9 | 78.6 | 88.5 | 83.7 | 88.6 | - |
| | JEDI | 1.39M | 7B | 96.9 | 87.2 | 95.9 | 87.9 | 94.4 | 84.2 | 91.7 | - |
| RL | UI-R1 | 136 | 3B | 95.6 | 84.7 | 85.2 | 73.3 | 90.2 | 59.3 | 83.3 | - |
| | GUI-R1 | 3K | 3B | 96.7 | 76.7 | 89.6 | 72.1 | 93.8 | 64.8 | 83.6 | - |
| | GUI-G1 | 17K | 3B | 98.6 | 85.8 | 96.4 | 80.7 | 91.4 | 82.3 | 90.3 | - |
| | GUI-G$^2$ | 100K | 7B | 96.7 | 90.8 | 95.9 | 88.6 | 90.9 | 86.9 | 92.0 | - |
| Attention-based / TF | TAG | 0 | 8.5B | 88.3 | 29.3 | 82.5 | 28.6 | 70.9 | 29.1 | 57.5 | - |
| | Trifuse | 0 | 3B | 91.9 | 73.4 | 90.2 | 65.7 | 87.4 | 70.4 | 81.1 | **+23.6** |
| | Trifuse | 0 | 7B | 92.7 | 79.0 | 90.2 | 80.0 | 91.7 | 79.6 | 86.2 | **+28.7** |
| TB | GUI-Actor | 1M | 3B | 93.0 | 79.9 | 88.1 | 78.6 | 90.9 | 84.0 | 86.5 | - |
| | Trifuse+GUI-Actor | - | 3B | 96.0 | 82.5 | 90.7 | 80.7 | 96.5 | 85.4 | 89.5 | **+3.0** |
| | GUI-Actor | 1M | 7B | 94.9 | 82.1 | 91.8 | 80.0 | 91.3 | 85.4 | 88.3 | - |
| | Trifuse+GUI-Actor | - | 7B | 97.8 | 84.3 | 90.7 | 82.9 | 95.7 | 86.9 | 90.5 | **+2.2** |
| | GUI-AIMA | 85K | 3B | 96.3 | 83.8 | 94.3 | 85.7 | 92.6 | 72.3 | 88.1 | - |
| | Trifuse+GUI-AIMA | - | 3B | 98.2 | 86.0 | 96.9 | 85.0 | 94.8 | 79.1 | 90.6 | **+2.5** |

*Table 3.* Performance comparison on ScreenSpot-v2.

| | Method | Training Data Size | Model Size | Mobile | | Desktop | | Web | | Avg. | Δ |
|---|---|---|---|---|---|---|---|---|---|---|---|
| | | | | Text | Icon | Text | Icon | Text | Icon | | |
| General | Qwen2.5-VL | - | 3B | 64.8 | 49.8 | 62.4 | 32.9 | 59.6 | 50.5 | 55.0 | - |
| | Qwen2.5-VL | - | 7B | 90.8 | 73.4 | 85.1 | 62.9 | 75.2 | 64.1 | 76.6 | - |
| | Operator | - | - | 48.3 | 41.5 | 90.2 | 80.3 | 92.8 | 84.3 | 70.5 | - |
| | OmniParser-v2 | - | - | 95.5 | 74.6 | 92.3 | 60.9 | 88.0 | 59.6 | 80.7 | - |
| SFT | UI-TARS | 50B Token | 2B | 95.2 | 79.1 | 90.7 | 68.6 | 87.2 | 78.3 | 84.7 | - |
| | UI-TARS | 50B Token | 7B | 96.9 | 89.1 | 95.4 | 85.0 | 93.6 | 85.2 | 91.6 | - |
| | JEDI | 1.39M | 3B | 96.9 | 81.5 | 96.9 | 78.6 | 88.5 | 83.7 | 88.6 | - |
| | JEDI | 1.39M | 7B | 96.9 | 87.2 | 95.9 | 87.9 | 94.4 | 84.2 | 91.7 | - |
| RL | UI-R1 | 136 | 3B | 96.2 | 84.3 | 92.3 | 63.6 | 89.2 | 75.4 | 85.4 | - |
| | GUI-R1 | 3K | 3B | 97.6 | 78.2 | 94.3 | 64.3 | 91.0 | 72.4 | 85.0 | - |
| | GUI-G$^2$ | 100K | 7B | 98.3 | 91.9 | 95.4 | 89.3 | 94.0 | 87.7 | 93.3 | - |
| Attention-based / TF | TAG | 0 | 8.5B | 78.8 | 24.0 | 72.2 | 25.0 | 67.4 | 25.8 | 51.2 | - |
| | Trifuse | 0 | 3B | 92.7 | 75.1 | 92.8 | 64.3 | 91.3 | 70.9 | 82.6 | **+31.4** |
| | Trifuse | 0 | 7B | 95.2 | 79.0 | 93.3 | 79.3 | 92.2 | 78.2 | 86.9 | **+35.7** |
| TB | GUI-Actor | 1M | 2B | 97.6 | 83.4 | 96.9 | 83.6 | 94.0 | 85.7 | 91.0 | - |
| | Trifuse+GUI-Actor | - | 2B | 97.8 | 84.3 | 96.4 | 85.7 | 94.8 | 86.9 | 91.5 | **+0.5** |
| | GUI-Actor | 1M | 7B | 97.6 | 88.2 | 96.9 | 85.7 | 93.2 | 86.7 | 92.1 | - |
| | Trifuse+GUI-Actor | - | 7B | 98.2 | 90.8 | 97.4 | 87.1 | 93.9 | 88.3 | 93.2 | **+1.1** |
| | GUI-AIMA | 85K | 3B | 99.2 | 85.9 | 96.1 | 88.9 | 96.1 | 80.2 | 91.5 | - |
| | Trifuse+GUI-AIMA | - | 3B | 98.2 | 87.3 | 96.9 | 90.7 | 97.0 | 85.4 | 92.9 | **+1.4** |

*Table 4.* Performance comparison on ScreenSpot-Pro.

| | Method | Training Data Size | Model Size | CAD | Dev | Creative | Scientific | Office | OS | Avg. | Δ |
|---|---|---|---|---|---|---|---|---|---|---|---|
| General | GPT-4o | - | / | 1.5 | 0.7 | 0.6 | 1.3 | 0.8 | 0.0 | 0.8 | - |
| | Qwen2.5-VL | - | 3B | 8.7 | 12.1 | 17.1 | 26.3 | 29.6 | 6.1 | 16.1 | - |
| | Qwen2.5-VL | - | 7B | 13.2 | 26.1 | 24.8 | 33.1 | 45.2 | 23.5 | 26.8 | - |
| | Claude | - | / | 11.9 | 13.2 | 17.1 | 26.9 | 26.9 | 8.0 | 17.1 | - |
| SFT | UGround-v1 | 1.3M | 7B | 12.3 | 28.1 | 32.6 | 41.0 | 49.6 | 24.5 | 31.1 | - |
| | UI-TARS | 50B Token | 72B | 17.3 | 40.7 | 40.7 | 47.8 | 54.8 | 30.1 | 38.1 | - |
| | JEDI | 1.39M | 3B | 23.1 | 38.1 | 35.8 | 40.3 | 57.0 | 25.0 | 36.1 | - |
| | JEDI | 1.39M | 7B | 31.6 | 36.1 | 41.3 | 49.5 | 65.7 | 33.2 | 42.0 | - |
| RL | UI-R1 | 136 | 3B | 10.0 | 13.7 | 17.9 | 30.6 | 27.4 | 9.2 | 17.8 | - |
| | GUI-G1 | 17K | 3B | 32.4 | 31.1 | 26.9 | 49.6 | 59.1 | 17.6 | 37.1 | - |
| | GUI-G$^2$ | 100K | 7B | 17.7 | 22.8 | 27.4 | 30.8 | 39.1 | 17.9 | 25.5 | - |
| Attention-based / TF | TAG | 0 | 8.5B | 5.4 | 2.0 | 2.9 | 2.6 | 2.6 | 2.0 | 3.0 | - |
| | Trifuse | 0 | 3B | 17.1 | 14.1 | 17.7 | 22.1 | 30.4 | 12.8 | 18.9 | **+15.9** |
| | Trifuse | 0 | 7B | 28.8 | 27.7 | 22.7 | 31.2 | 47.8 | 26.5 | 29.7 | **+26.7** |
| TB | GUI-Actor | 1M | 3B | 34.1 | 39.8 | 36.7 | 49.6 | 61.3 | 35.2 | 42.2 | - |
| | Trifuse+GUI-Actor | - | 3B | 35.2 | 41.1 | 37.5 | 48.6 | 63.4 | 34.7 | 42.7 | **+0.5** |
| | GUI-Actor | 1M | 7B | 38.3 | 38.1 | 41.4 | 50.8 | 63.0 | 38.8 | 44.6 | - |
| | Trifuse+GUI-Actor | - | 7B | 40.0 | 40.5 | 42.0 | 53.3 | 61.7 | 40.8 | 45.8 | **+1.2** |
| | GUI-AIMA | 85K | 3B | 39.3 | 48.9 | 44.7 | 57.0 | 65.6 | 50.0 | 49.8 | - |
| | Trifuse+GUI-AIMA | - | 3B | 42.0 | 51.2 | 45.4 | 56.2 | 67.8 | 51.5 | 51.3 | **+1.5** |

global maxima across different approaches.

Several key observations can be drawn from the results. (1) Despite requiring no task-specific fine-tuning, Trifuse achieves competitive or superior performance compared to supervised fine-tuned (SFT) models and even outperforms several reinforcement learning (RL)-based methods across all benchmarks. (2) Using a 3B-parameter backbone, Trifuse attains higher average accuracy than existing training-free attention-based methods, substantially surpassing the TAG baseline (Xu et al., 2025). (3) When integrated with existing attention-based methods such as GUI-Actor and GUI-AIMA (Wu et al., 2025b; Zhou et al., 2025a), Trifuse consistently improves grounding accuracy, demonstrating its effectiveness as a modular enhancement.

Overall, these results validate that the proposed Consensus-SinglePeak (CS) fusion strategy effectively captures complementary information from attention, OCR, and caption modalities. Trifuse delivers robust performance gains across diverse platforms, resolutions, and element types, while remaining entirely independent of GUI-specific annotated data.

### 4.2. Ablation Studies

We perform systematic ablation studies to examine the contribution of the core components of Trifuse, including the attention extraction strategy (token and head-level filtering) and the proposed Consensus-SinglePeak (CS) fusion strategy. These studies are designed to isolate the effect of each component and to assess their individual and combined impact on grounding performance. Unless otherwise specified,

all ablation experiments adopt Qwen2.5-VL-3B-Instruct as the backbone. Additional experimental results, including sensitivity analyses of key hyperparameters, efficiency analysis and ablation studies on two-stage localization strategy are reported in Appendix B.

**Attention Modality** We analyze the effect of token filtering and head filtering on grounding performance when using the attention modality alone. Table 6 reports the results under different selection strategies.

**Head filtering.** With a fixed token selection strategy, head filtering has a pronounced impact on grounding accuracy. Aggregating attention from all heads leads to substantial performance degradation, indicating that many attention heads encode information that is irrelevant to spatial localization. Restricting attention to a subset of heads (layer 16 to layer 32) yields moderate improvements even without explicit ranking. In contrast, our proposed top-$k$ head selection strategy, which ranks heads by spatial entropy, consistently achieves the best performance by retaining only the most spatially informative heads.

*Table 5.* Performance comparison on OSWorld-G.

| | Model | Training Data Size | Model Size | Text Matching | Element Recognition | Layout Understanding | Fine-grained Manipulation | Avg. | Δ |
|---|---|---|---|---|---|---|---|---|---|
| General | Operator | - | - | 51.3 | 42.4 | 46.6 | 31.5 | 40.6 | - |
| | Gemini-2.5-Pro | - | - | 59.8 | 45.5 | 49.0 | 33.6 | 45.2 | - |
| | Qwen2.5-VL | - | 3B | 41.4 | 28.8 | 34.8 | 13.4 | 27.3 | - |
| | Qwen2.5-VL | - | 7B | 45.6 | 32.7 | 41.9 | 18.1 | 31.4 | - |
| SFT | UGround-V1 | 1.3M | 7B | 51.3 | 40.3 | 43.5 | 24.8 | 36.4 | - |
| | UI-TARS | 50B Token | 7B | 60.2 | 51.8 | 54.9 | 35.6 | 47.5 | - |
| | JEDI | 1.39M | 3B | 67.4 | 53.0 | 53.8 | 44.3 | 50.9 | - |
| | JEDI | 1.39M | 7B | 65.9 | 55.5 | 57.7 | 46.9 | 54.1 | - |
| Attention-based — TF | TAG | 0 | 8.5B | 44.9 | 19.2 | 28.1 | 8.4 | 25.3 | - |
| | Trifuse | 0 | 3B | 55.2 | 33.0 | 45.6 | 24.7 | 38.7 | **+13.4** |
| | Trifuse | 0 | 7B | 65.3 | 35.9 | 50.4 | 27.2 | 43.6 | **+18.3** |
| Attention-based — TB | GUI-Actor | 1M | 3B | 64.4 | 60.6 | 64.8 | 33.6 | 54.6 | - |
| | Trifuse+GUI-Actor | - | 3B | 64.9 | 61.4 | 66.7 | 34.4 | 56.5 | **+1.9** |
| | GUI-Actor | 1M | 7B | 65.9 | 62.7 | 66.4 | 38.2 | 56.6 | - |
| | Trifuse+GUI-Actor | - | 7B | 66.4 | 62.3 | 67.1 | 38.9 | 57.9 | **+1.3** |
| | GUI-AIMA | 85K | 3B | 64.8 | 65.5 | 68.8 | 36.8 | 58.3 | - |
| | Trifuse+GUI-AIMA | - | 3B | 65.6 | 63.7 | 69.4 | 36.3 | 58.4 | **+0.1** |

*Table 6.* Ablation studies on token and head selection strategies for attention extraction.

| | Method | ScreenSpot Text | Icon | Avg. | Δ | ScreenSpot-v2 Text | Icon | Avg. | Δ | ScreenSpot-Pro Avg. | Δ | OSWorld-G Avg. | Δ |
|---|---|---|---|---|---|---|---|---|---|---|---|---|---|
| Top Token | All Head | 13.2 | 6.6 | 10.2 | - | 11.5 | 6.2 | 9.2 | - | 3.6 | - | 7.3 | - |
| | Range Head | 18.4 | 10.8 | 14.9 | +4.7 | 17.0 | 9.8 | 13.8 | +4.6 | 6.2 | +1.6 | 14.9 | +7.6 |
| | Top Head | 66.5 | 48.7 | 58.5 | +48.3 | 62.2 | 45.1 | 54.7 | +45.5 | 8.0 | +4.4 | 28.0 | +20.7 |
| Top Head | All Token | 50.9 | 32.3 | 42.5 | - | 49.9 | 29.9 | 40.9 | - | 5.8 | - | 18.6 | - |
| | Last Token | 61.6 | 44.2 | 53.7 | +11.2 | 60.6 | 41.1 | 52.1 | +11.2 | 7.3 | +1.5 | 23.7 | +5.1 |
| | Top Token | 66.5 | 48.7 | 58.5 | +16.0 | 62.2 | 45.1 | 54.7 | +13.8 | 8.0 | +2.2 | 28.0 | +9.4 |

**Token filtering.** With a fixed head selection strategy, token filtering also plays a critical role. Aggregating attention across all instruction tokens introduces noise from task-irrelevant words, which dilutes the spatial grounding signal. Using only the final token partially alleviates this issue but may discard important semantic cues related to the target element. Our top-$k$ token selection strategy, which ranks tokens according to their visual relevance scores, achieves the strongest performance by focusing on semantically informative tokens.

Overall, these results demonstrate that selective attention extraction through joint filtering of both tokens and heads is crucial for effective attention-based GUI grounding.

**Fusion Strategy** Table 7 reports ablation results for different fusion strategies. We analyze the performance of individual modalities to reveal their complementary strengths and limitations, and then compare alternative fusion strategies.

**Single-modality analysis.** Each modality exhibits distinct and complementary characteristics. The OCR modality achieves high accuracy on text-based elements but performs poorly on icon elements, reflecting its strong capability in text localization and its inherent limitation on purely visual targets. In contrast, the caption modality provides more balanced performance by introducing semantic descriptions of visual elements; however, its overall accuracy remains limited due to imperfect caption quality and semantic matching. The attention modality yields the strongest single-modality performance by leveraging the MLLM's visual ability, yet it still falls short of state-of-the-art results when used in isolation. These observations highlight the complementary nature of the three modalities: OCR excels at textual cues, captions capture visual semantics, and attention offers broad but coarse coverage.

*Table 7.* Ablation studies on fusion strategies. We compare single modality (attention, OCR, caption), simple averaging, weighted fusion, and our proposed cross-modal consistency (CS) fusion.

| | Method | ScreenSpot Text | Icon | Avg. | Δ | ScreenSpot-v2 Text | Icon | Avg. | Δ | ScreenSpot-Pro Avg. | Δ | OSWorld-G Avg. | Δ |
|---|---|---|---|---|---|---|---|---|---|---|---|---|---|
| Single | OCR | 36.3 | 5.0 | 22.1 | - | 31.9 | 3.2 | 19.7 | - | 4.9 | - | 14.3 | - |
| | Caption | 29.8 | 22.9 | 26.7 | - | 29.0 | 16.2 | 23.2 | - | 3.7 | - | 10.2 | - |
| | Attention | 66.5 | 48.7 | 58.5 | - | 62.2 | 45.1 | 54.7 | - | 8.0 | - | 28.0 | - |
| Fusion | Average | 78.5 | 43.4 | 63.3 | - | 83.3 | 44.5 | 65.8 | - | 13.5 | - | 33.6 | |
| | Custom | 63.2 | 51.2 | 58.0 | -5.5 | 78.7 | 42.1 | 62.2 | -3.6 | 10.7 | -2.8 | 29.1 | -4.5 |
| | CS (Trifuse) | 89.9 | 70.5 | 81.1 | +17.8 | 92.3 | 71.0 | 82.6 | +16.8 | 18.9 | +5.4 | 43.6 | +10.0 |

**Fusion mechanism comparison.** We compare three fusion strategies. The Average baseline uniformly combines the three modalities, achieving modest improvements by leveraging complementary information. However, treating all modalities equally limits its effectiveness, as the relative importance of each modality varies across task types. The Custom baseline applies a fixed, manually specified weighting scheme to fuse modality-specific heatmaps, assigning weights of 0.6 to the attention modality and 0.2 to both the OCR and caption modalities across all benchmarks. Despite using the same weights in all settings, this baseline exhibits inconsistent performance, indicating that fixed fusion weights are insufficient to adapt to the diverse characteristics of GUI grounding tasks.

In contrast, CS fusion strategy achieves substantial and consistent improvements across all benchmarks. Its effectiveness arises from two complementary mechanisms: (1) the consensus term $\mathbf{M}^{cons}$ amplifies regions where multiple modalities agree, providing robustness against modality-specific noise, and (2) the single-peak term $\mathbf{M}^{single}$ preserves strong discriminative signals from individual modalities, enabling adaptation to task-specific characteristics. This adaptive weighting enables CS fusion to substantially outperform both single modalities and alternative fusion strategies, validating its effectiveness for GUI grounding.

# 5. Conclusion

We introduce Trifuse, an attention-based framework for GUI grounding that leverages complementary spatial cues through principled fusion without task-specific fine-tuning. By systematically fusing attention, OCR-derived textual cues, and icon-level caption semantics through the proposed Consensus-SinglePeak (CS) mechanism, Trifuse substantially improves grounding performance over existing attention-based approaches, while approaching the performance of supervised fine-tuned methods and reinforcement

learning-based methods without relying on any GUI-specific training data. Extensive ablation studies validate the effectiveness of Trifuse's core design choices, including (1) token filtering based on visual relevance scores to identify semantically informative query terms, (2) head filtering via spatial entropy to retain spatially focused attention heads, and (3) CS fusion, which adaptively balances cross-modal consensus with modality-specific discriminative peaks. Together with the proposed two-stage localization strategy, Trifuse achieves strong and robust grounding performance. Overall, our results demonstrate that principled multimodal fusion at inference time can enable strong GUI grounding capabilities from MLLMs, highlighting a promising direction for scalable, data-efficient GUI agents.

## Acknowledgments

This work was supported by the National Natural Science Foundation of China (project no. 62406329, 62476280).

## Impact Statement

This paper presents work whose goal is to advance the field of machine learning by improving the reliability and generalization of graphical user interface (GUI) grounding for autonomous GUI agents. Improved GUI grounding may benefit applications such as assistive technologies, automated software testing, and human–computer interaction by enabling more accurate interpretation of diverse interfaces.

The methods studied in this work could potentially be misused to automate interactions with software systems in unintended ways if deployed without appropriate safeguards, a risk common to GUI automation techniques in general. We believe the primary contribution of this work is methodological, providing a foundational improvement to GUI perception rather than enabling applications with direct societal risks.

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

*Table 8.* Performance comparison on UI-Vision across element grounding and layout grounding tasks.

| Method | Model Size | Element | | | Layout | | |
|---|---|---|---|---|---|---|---|
| | | Basic | Functional | Spatial | IOU | Precision | Recall |
| GPT-4o | - | 1.6 | 1.5 | 1.0 | 20.0 | 59.6 | 24.1 |
| Gemini-1.5-pro | - | 0.8 | 0.3 | 0.6 | **30.8** | **67.8** | 36.9 |
| Claude-3.7-Sonnet | - | 9.5 | 7.7 | **7.6** | 17.6 | 31.5 | 34.1 |
| Qwen2.5-VL | 3B | 1.0 | 0.1 | 0.5 | 5.0 | 52.9 | 5.8 |
| Qwen2.5-VL | 7B | 1.2 | 0.8 | 0.5 | 5.6 | 62.1 | 6.4 |
| CogAgent | 9B | 12.0 | **12.2** | 2.6 | 6.2 | 8.0 | **42.9** |
| SeeClick | 9.6B | 9.4 | 4.7 | 2.1 | 5.1 | 6.3 | 30.1 |
| OSAtlas | 7B | 12.2 | 11.2 | 3.7 | 28.2 | 66.4 | 41.6 |
| TAG | 8.5B | 10.3 | 8.0 | 1.7 | 1.5 | 42.8 | 1.6 |
| Trifuse | 3B | 12.4 | 9.7 | 4.8 | 5.8 | 58.1 | 6.8 |
| Trifuse | 7B | **13.6** | 10.3 | 5.8 | 6.0 | 62.6 | 7.1 |

# A. Benchmark Details

**ScreenSpot** (Cheng et al., 2024) is a widely-used zero-shot GUI grounding benchmark spanning desktop, mobile, and web interfaces. It provides separate evaluations for text-based and icon-based (widget-level) localization tasks, enabling fine-grained analysis of grounding capabilities across different element types.

**ScreenSpot-v2** (Cheng et al., 2024) addresses critical quality issues in the original ScreenSpot dataset, provides cleaned annotations with verified element references and unambiguous instructions, enabling more reliable evaluation.

**ScreenSpot-Pro** (Li et al., 2025) extends the evaluation to high-resolution scenarios with more complex interface layouts and fine-grained element localization challenges, testing model robustness on realistic production-scale GUIs.

**OSWorld-G** (Xie et al., 2025) comprises 564 meticulously annotated samples designed to systematically evaluate diverse grounding capabilities: text matching, element recognition, layout understanding, fine-grained manipulation, and infeasibility detection (identifying when target elements are absent). Each sample includes element-type annotations, providing diagnostic insights into model performance across different GUI component categories.

**ScreenSpot**, **ScreenSpot-v2**, and **ScreenSpot-Pro** categorize tasks into two types: text and icon. **OSWorld-G** further categorizes tasks into five types: Text Matching, Element Recognition, Layout Understanding, Fine-grained Manipulation, and Refusal. The Refusal category corresponds to cases where the target element specified by the instruction does not appear in the screenshot, and the model is required to output a null prediction (i.e., $(-1, -1)$). This categorization also explains why Trifuse does not achieve substantial improvements on OSWorld-G, as our method is designed for grounding visible GUI elements and does not explicitly handle the Refusal task category.

# B. Additional Experiments

## B.1. Addition Experiments on UI-Vision

To further validate the generalization of Trifuse, we conduct additional experiments on **UI-Vision** (Nayak et al., 2025), a desktop-focused high-resolution benchmark that features denser layouts, smaller UI elements, and higher native resolutions than the benchmarks in our main evaluation. These properties place greater stress on both the attention extraction pipeline and the two-stage localization component, providing a complementary testbed for assessing robustness.

Results are presented in Table 8. Trifuse demonstrates strong competitiveness on UI-Vision: across all three element grounding subtasks, both Trifuse variants consistently outperform Qwen2.5-VL backbone and TAG. This confirms that the CS fusion mechanism and the two-stage zoom-in localization strategy remain effective in challenging high-resolution desktop scenarios characterized by dense layouts and small UI elements.

Regarding the layout grounding subtasks, UI-Vision evaluates a model's ability to cluster UI elements into functional and semantic groups and predict bounding boxes that encapsulate them. Since Trifuse is designed for point-wise grounding, its comparatively lower performance on this subtask is expected and consistent with its design objective. We regard this

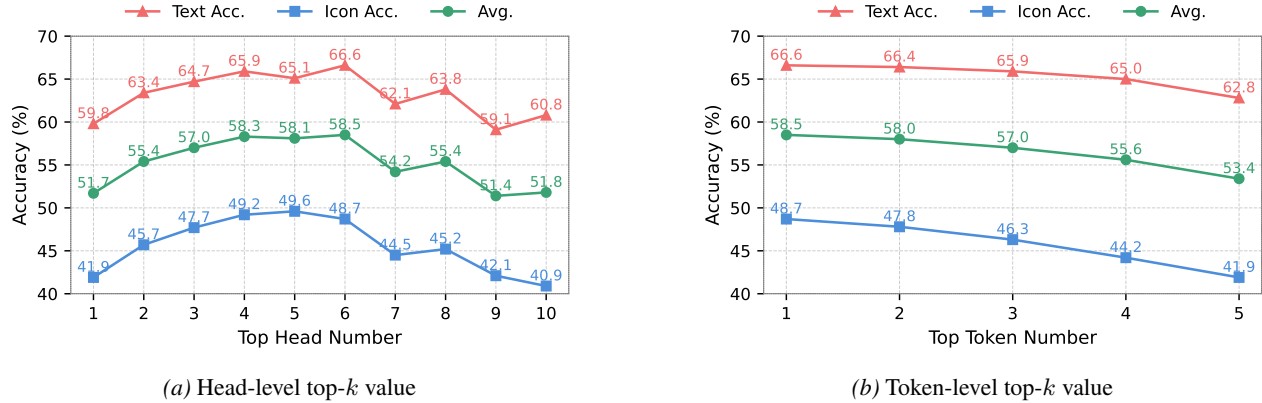

*(a)* Head-level top-$k$ value        *(b)* Token-level top-$k$ value

*Figure 3.* Effect of the number of Top-k Head value and Top-k token value for Attention Extraction on ScreenSpot

as a meaningful direction for future work, where extending Trifuse to support region-level predictions could broaden its applicability to layout-aware GUI understanding tasks.

### B.2. Hyperparameter Analysis

Trifuse contains several tunable hyperparameters: the top-$k$ token and top-$k$ head numbers for attention extraction, the modality-wise peak threshold $\tau_s$, and the fusion coefficients $\lambda$, $\alpha$, and $\beta$. We conduct sensitivity analyses on ScreenSpot to characterize the influence of each hyperparameter.

**Top-$k$ Head and Token Selection.** Figure 3 and Tables 9–10 report the effect of varying top-$k$ value for head and token selection, evaluated on the attention modality alone and Trifuse-3B.

For head selection (Table 9), performance exhibits a non-monotonic relationship with top-$k$ value: too few heads fail to capture sufficient localization signals, while too many introduce heads weakly correlated with grounding, reducing effectiveness. Empirically, a moderate top-$k$ head number achieves the best balance between signal coverage and noise reduction.

For token selection (Table 10), setting $k{=}1$ yields the best performance for both attention-only and Trifuse-3B. Increasing $k$ monotonically degrades accuracy, as additional tokens introduce semantically irrelevant noise that dilutes the grounding signal.

Based on these results, we set $k{=}1$ for tokens and $k{=}6$ for heads in all experiments.

**Fusion Hyperparameters.** Tables 11–13 report the sensitivity of Trifuse to the fusion coefficients $\tau_s$, $\lambda$, and $(\alpha, \beta)$.

For the peak threshold $\tau_s$ (Table 11), performance across all configurations varies smoothly. Trifuse-3B achieves its best result of 80.8% at $\tau_s{=}0.8$, with a total performance range of 1.8% across the evaluated values ($[0.65, 0.90]$). Consistent trends are observed for Trifuse-7B and Trifuse+GUI-Actor/AIMA variants, all peaking near $\tau_s \in \{0.75, 0.80\}$.

For the peak amplification coefficient $\lambda$ (Table 12), Trifuse-3B peaks at $\lambda{=}0.5$ (81.1%) with a performance range of only 0.9% across $\lambda \in [0.3, 0.7]$, indicating low sensitivity. Similar stability is observed across all model variants, with optimal values consistently in the range $\lambda \in \{0.5, 0.6\}$.

For the confidence estimation parameters $(\alpha, \beta)$ (Table 13), the joint grid search on Trifuse-3B shows a peak of 81.1% at $(\alpha{=}10, \beta{=}2.0)$, with a total variation of 1.2% across all evaluated combinations. Performance is most sensitive to $\alpha$, with mid-range values ($\alpha \in \{8, 10, 12\}$) consistently outperforming the extremes. Based on these analyses, we fix $\lambda{=}0.5$, $\alpha{=}10$, and $\beta{=}2.0$ across all experiments.

Notably, even under the least favorable hyperparameter configurations, Trifuse-3B (77.7%) substantially outperforms the TAG baseline (57.5%), confirming that the performance gains of Trifuse are robust and not contingent on precise hyperparameter tuning.

*Table 9.* Effect of the number of Top-$k$ Head value for Attention Extraction on ScreenSpot (accuracy, %).

| Method | Model Size | Top-$k$ Head | | | | | | | | | |
|---|---|---|---|---|---|---|---|---|---|---|---|
| | | $k$=1 | $k$=2 | $k$=3 | $k$=4 | $k$=5 | $k$=6 | $k$=7 | $k$=8 | $k$=9 | $k$=10 |
| Only Attention | – | 51.7 | 55.4 | 57.0 | 58.3 | 58.1 | **58.5** | 54.2 | 55.4 | 51.4 | 51.8 |
| Trifuse | 3B | 78.1 | 78.8 | 79.5 | **81.1** | 80.8 | **81.1** | 80.5 | 80.9 | 78.9 | 79.2 |

*Table 10.* Effect of the number of Top-$k$ Token value for Attention Extraction on ScreenSpot (accuracy, %).

| Method | Model Size | Top-$k$ Token | | | | |
|---|---|---|---|---|---|---|
| | | $k$=1 | $k$=2 | $k$=3 | $k$=4 | $k$=5 |
| Only Attention | – | **58.5** | 58.0 | 57.0 | 55.6 | 53.4 |
| Trifuse | 3B | **81.1** | 80.5 | 79.2 | 78.2 | 77.7 |

### B.3. Efficiency Analysis

We conduct a dedicated efficiency evaluation on A800-80G. For each benchmark, we randomly sample 20 examples, repeat each measurement 10 times, and report the average wall-clock inference time in Tables 14 and 15. A component-wise latency breakdown is provided in Table 16.

**End-to-end latency.** As shown in Table 14, Trifuse incurs approximately 7–10 seconds of end-to-end latency per image, compared to ∼0.5s for the Qwen2.5-VL backbone alone. Relative to TAG (4–6s), Trifuse adds a moderate overhead while delivering substantially higher accuracy. Table 15 highlights this accuracy–efficiency tradeoff.

**Component-wise breakdown.** Table 16 reveals that the icon caption model accounts for 83–88% of total inference time across all Trifuse variants, constituting the dominant latency bottleneck. In contrast, the CS fusion strategy itself contributes negligible overhead ($< 0.01\%$), and MLLM inference, attention extraction, and OCR together account for less than 16% of total latency. These results confirm that the proposed fusion mechanism and attention extraction pipeline are computationally lightweight; the primary overhead stems from the caption model rather than the core Trifuse components. Replacing the icon caption model with a more lightweight yet accurate alternative represents the most promising future direction for reducing end-to-end latency.

### B.4. Additional Ablation Studies

**Ablation Study of Two-Stage Localization** Table 17 compares our two-stage localization strategy against direct coordinate prediction from the fused heatmap. Trifuse consistently outperforms the direct baseline across all benchmarks. The two-stage approach effectively increases spatial resolution for precise localization, particularly benefiting small GUI elements on ScreenSpot-Pro where direct prediction from downsampled images struggles. Notably, even the direct localization variant of Trifuse surpasses the TAG (Xu et al., 2025) baseline on all benchmarks, further validating the effectiveness of the overall Trifuse framework.

**Detailed Main Ablation Studies** Table 18, Table 19 and Table 20 presents detailed ablation results on ScreenSpot and ScreenSpot-v2, breaking down performance by element type (text vs. icon) across platforms using Qwen2.5-VL-3B-Instruct as the base model. Performance improvements are observed across both element types, further confirming the effectiveness of fusion strategy, fine-grained token and head selection.

## C. Limitations

While Trifuse demonstrates strong performance for GUI grounding, several limitations suggest directions for future work.

**Caption Model Quality** Trifuse's overall performance depends on the quality of the OCR and caption detection models. While current OCR engines (e.g., PaddleOCR v4 (Cui et al., 2025)) achieve high accuracy on text detection, existing icon caption models show limited performance. The lack of lightweight, high-quality, efficiency GUI icon detection models constrains Trifuse's potential. Future work could explore improved icon captioning models or alternative visual modalities

*Table 11.* Effect of $\tau_s$ on ScreenSpot (accuracy, %).

| Method | Model Size | $\tau_s$ | | | | | |
|---|---|---|---|---|---|---|---|
| | | 0.65 | 0.70 | 0.75 | 0.80 | 0.85 | 0.90 |
| Trifuse | 3B | 79.0 | 79.8 | 80.2 | **80.8** | 80.3 | 79.6 |
| Trifuse | 7B | 84.0 | 84.7 | 85.3 | **85.7** | 85.2 | 84.7 |
| Trifuse+GUI-Actor | 3B | 88.0 | 89.1 | **89.5** | 89.2 | 89.0 | 88.4 |
| Trifuse+GUI-Actor | 7B | 88.8 | 89.4 | 89.8 | **90.6** | 90.3 | 89.6 |
| Trifuse+GUI-AIMA | 3B | 88.4 | 89.2 | 89.6 | **90.5** | 90.3 | 89.7 |

*Table 12.* Effect of $\lambda$ on ScreenSpot (accuracy, %).

| Method | Model Size | $\lambda$ | | | | |
|---|---|---|---|---|---|---|
| | | 0.3 | 0.4 | 0.5 | 0.6 | 0.7 |
| Trifuse | 3B | 80.2 | 80.9 | **81.1** | 81.0 | 80.8 |
| Trifuse | 7B | 85.3 | 85.5 | **86.2** | 86.1 | 85.7 |
| Trifuse+GUI-Actor | 3B | 88.7 | 89.0 | 89.5 | **89.6** | 89.2 |
| Trifuse+GUI-Actor | 7B | 89.6 | 90.0 | 90.5 | **90.6** | 90.3 |
| Trifuse+GUI-AIMA | 3B | 89.9 | 90.4 | **90.6** | 90.4 | 89.4 |

to address this bottleneck.

**Performance Ceiling** While Trifuse substantially outperforms prior attention-based methods, its performance is fundamentally constrained by the capabilities of the underlying pretrained models. Scaling to larger or more specialized backbones yields diminishing returns, suggesting inherent limitations in training-free approaches compared to GUI-specific fine-tuning. Hybrid methods that combine Trifuse's multimodal fusion with lightweight task-specific fine-tuning represent a promising direction for future work.

## D. Prompt

For ScreenSpot, ScreenSpot-v2 and ScreenSpot-Pro, Trifuse, GUI-Actor, and GUI-AIMA all use models from the Qwen2.5-VL series, requiring only configuration of the system prompt. Figure 4 shows the prompt template used across all methods. For OSWorld-G, we adopt the prompt from the original benchmark in Figure 5 (Xie et al., 2025). Unlike other benchmarks, OSWorld-G allows models to return (-1, -1) coordinates when no target element is found. This abstention mechanism partly explains why Trifuse shows smaller improvements on OSWorld-G compared to other benchmarks, as Trifuse does not support this output format and must always produce coordinate predictions from fused heatmap.

## E. Case Study

### E.1. CS Fusion

To illustrate the CS fusion strategy, we present representative examples from ScreenSpot using Qwen2.5-VL-3B-Instruct with the same modality acquisition and modality fusion described in Section 3.

**Consensus**. Figure 6 shows a text-based grounding task on mobile platform. All three modalities—attention, OCR, and

> The system prompt for ScreenSpot, ScreenSpot-v2 and ScreenSpot-Pro evaluate
>
> You are a GUI-grounding assistant. Given a GUI screenshot, your task is to identify the region that best matches the user instruction.

*Figure 4.* System prompt used for ScreenSpot, ScreenSpot-v2, and ScreenSpot-Pro evaluation.

*Table 13.* Effect of $\alpha$ and $\beta$ on ScreenSpot of Trifuse-3B (accuracy, %).

|  | $\beta = 1.0$ | $\beta = 1.5$ | $\beta = 2.0$ | $\beta = 2.5$ | $\beta = 3.0$ |
|---|---|---|---|---|---|
| $\alpha = 6$ | 80.0 | 80.1 | 80.2 | 80.0 | 79.9 |
| $\alpha = 8$ | 80.2 | 80.8 | 80.7 | 80.7 | 80.3 |
| $\alpha = 10$ | 80.7 | 80.9 | **81.1** | 80.9 | 80.5 |
| $\alpha = 12$ | 80.5 | 80.7 | 80.8 | 80.7 | 80.3 |
| $\alpha = 14$ | 80.3 | 80.3 | 80.5 | 80.3 | 79.9 |

*Table 14.* Per-image inference time (seconds, mean $\pm$ std) of Trifuse and baseline methods across four benchmarks.

| Method | Model Size | ScreenSpot | ScreenSpot-v2 | ScreenSpot-Pro | OSWorld-G |
|---|---|---|---|---|---|
| MiniCPM-Llama3-V-2.5 | 8.5B | $0.79 \pm 0.06$ | $0.76 \pm 0.04$ | $0.80 \pm 0.08$ | $0.79 \pm 0.08$ |
| TAG | 8.5B | $4.00 \pm 1.04$ | $4.11 \pm 0.76$ | $4.55 \pm 1.84$ | $6.21 \pm 3.99$ |
| Qwen2.5-VL | 3B | $0.54 \pm 0.26$ | $0.75 \pm 0.19$ | $0.84 \pm 0.46$ | $0.35 \pm 0.13$ |
| Trifuse | 3B | $7.50 \pm 0.67$ | $7.78 \pm 0.70$ | $8.94 \pm 0.94$ | $8.40 \pm 1.15$ |
| Qwen2.5-VL | 7B | $0.59 \pm 0.29$ | $0.78 \pm 0.19$ | $0.78 \pm 0.36$ | $0.32 \pm 0.13$ |
| Trifuse | 7B | $7.79 \pm 0.99$ | $7.86 \pm 0.80$ | $9.99 \pm 1.39$ | $8.13 \pm 0.96$ |
| GUI-Actor | 3B | $0.52 \pm 0.24$ | $0.66 \pm 0.15$ | $0.75 \pm 0.41$ | $0.35 \pm 0.19$ |
| Trifuse+GUI-Actor | 3B | $7.64 \pm 0.98$ | $7.58 \pm 0.69$ | $11.96 \pm 2.68$ | $8.11 \pm 0.76$ |
| GUI-Actor | 7B | $0.51 \pm 0.22$ | $0.69 \pm 0.17$ | $0.78 \pm 0.33$ | $0.35 \pm 0.15$ |
| Trifuse+GUI-Actor | 7B | $6.58 \pm 0.64$ | $6.36 \pm 0.26$ | $8.16 \pm 2.29$ | $6.13 \pm 0.63$ |
| GUI-AIMA | 3B | $0.53 \pm 0.25$ | $0.69 \pm 0.19$ | $0.76 \pm 0.44$ | $0.35 \pm 0.16$ |
| Trifuse+GUI-AIMA | 3B | $7.38 \pm 0.94$ | $7.77 \pm 0.84$ | $9.22 \pm 0.77$ | $6.51 \pm 0.43$ |

caption—successfully localize the target region. CS fusion produces a refined heatmap that accurately integrates these consistent signals, resulting in precise localization.

**Single-peak**. When modalities disagree, CS fusion selectively amplifies the most confident prediction. In Figure 8 (icon task, mobile), only the caption modality localizes the target correctly while attention and OCR fail. CS fusion identifies this single strong signal and propagates it to the final heatmap. Similarly, in Figure 8 (icon task, desktop), only the attention modality succeeds. CS fusion again selectively incorporates the single-peak signal, demonstrating robustness when individual modalities provide complementary but sparse information.

These cases illustrate two key properties of CS fusion: (1) when modalities agree, it reinforces consensus regions; (2) when they disagree, it selectively trusts the most confident prediction rather than averaging conflicting signals.

### E.2. Head filtering on Attention modality

To motivate the necessity of head filtering, Figure 9 visualizes the attention maps of all 16 heads in layer 23 on a ScreenSpot example. The visualization reveals substantial heterogeneity across attention heads: only a small subset exhibit strong spatial alignment with the target element, while the majority of heads attend to irrelevant regions or produce diffuse attention patterns. This phenomenon is consistent across layers, where heads exhibit varying degrees of localization capability. These observations empirically motivate our head selection strategy—naive averaging across all heads would introduce significant noise from non-informative attention patterns, degrading grounding performance. By selectively aggregating attention only from heads with strong grounding signals, our approach effectively filters out this noise and enhances grounding performance.

The system prompt for OS-World-G evaluate

You are a helpful assistant.

# Tools
You may call one or more functions to assist with the user query.
You are provided with function signatures within <tools></tools> XML tags:
<tools>
{tool_descs}
</tools>

For each function call, return a json object with function name and arguments within <tool_call></tool_call> XML tags:
<tool_call>
{{"name": <function-name>, "arguments": <args-json-object>}}
</tool_call>

Additionally, if you think the task is infeasible (e.g., the task is not related to the image), return <tool_call>\n{{\"name\": \"computer_use\", \"arguments\": {{\"action\": \"wait\", \"time\": 10}}}}\n</tool_call>

*Figure 5.* System prompt used for OSWorld-G evaluation.

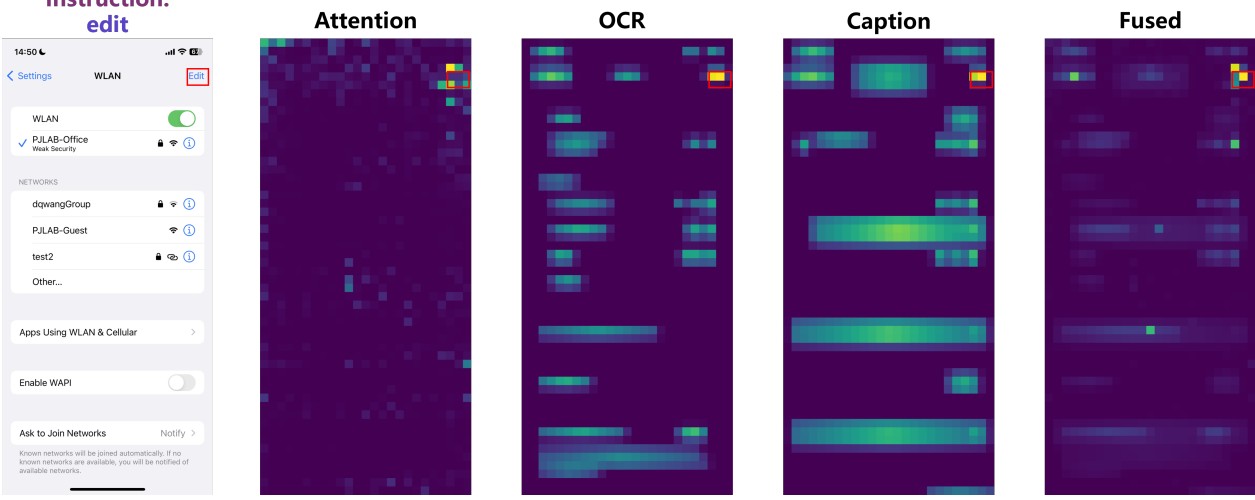

*Figure 6.* Visualization of CS fusion. All three modalities correctly localize the target, and fusion reinforces their agreement.

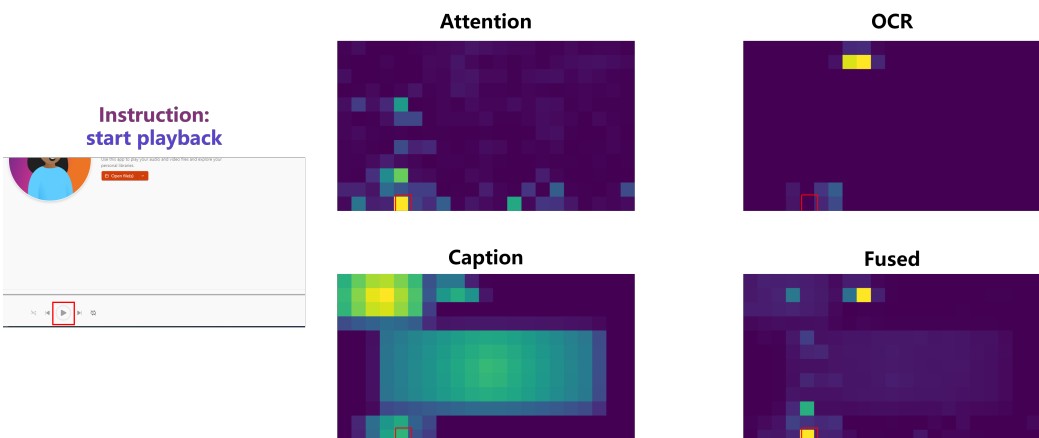

*Figure 7.* Visualization of CS fusion. Only attention modality correctly localizes the target; CS fusion selectively trusts the confident prediction.

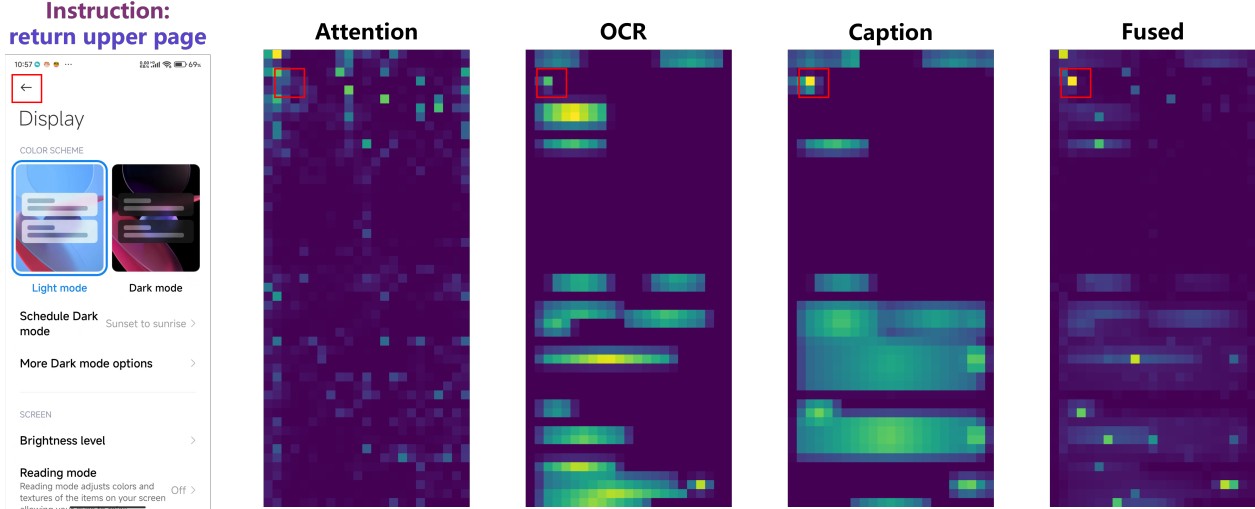

*Figure 8.* Visualization of CS fusion. Only caption modality correctly localizes the target; CS fusion selectively trusts the confident prediction.

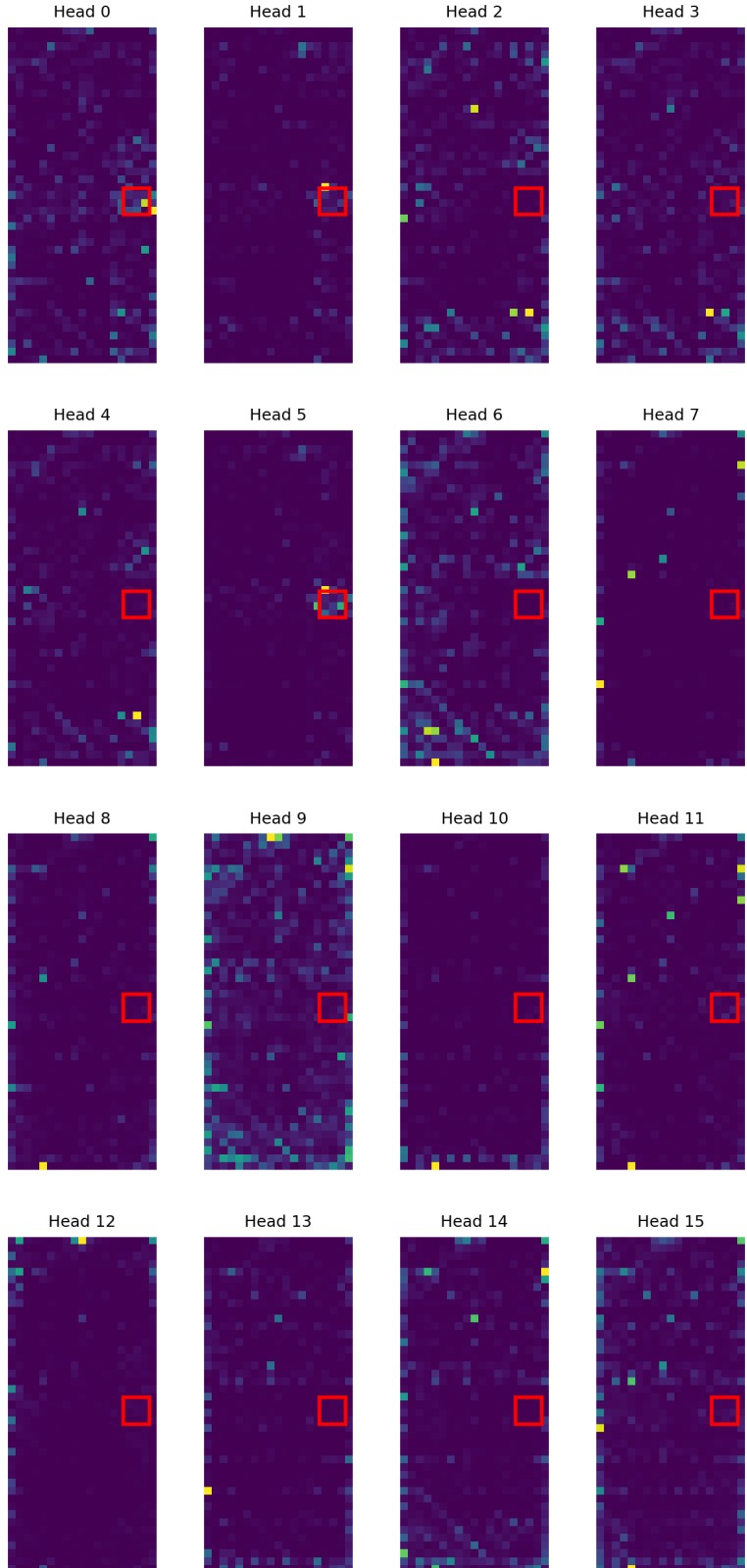

*Figure 9.* Visualization of attention maps across all 16 heads at layer 23 for a ScreenSpot example. The red bounding box marks the ground truth target element.

*Table 15.* Per-image inference time (seconds, mean ± std) and accuracy (%) of training-free attention-based methods across four benchmarks.

| Method | Size | ScreenSpot | | ScreenSpot-v2 | | ScreenSpot-Pro | | OSWorld-G | |
|---|---|---|---|---|---|---|---|---|---|
| | | Time ↓ | Acc. ↑ | Time ↓ | Acc. ↑ | Time ↓ | Acc. ↑ | Time ↓ | Acc. ↑ |
| TAG | 8.5B | 4.00 ± 1.04 | 57.5 | 4.11 ± 0.76 | 51.2 | 4.55 ± 1.84 | 3.0 | 6.21 ± 3.99 | 25.3 |
| Trifuse (3B) | 3B | 7.50 ± 0.67 | 81.1 | 7.78 ± 0.70 | 82.6 | 8.94 ± 0.94 | 18.9 | 8.40 ± 1.15 | 38.7 |
| Trifuse (7B) | 7B | 7.79 ± 0.99 | **86.2** | 7.86 ± 0.80 | **86.9** | 9.99 ± 1.39 | **29.7** | 8.13 ± 0.96 | **47.6** |

*Table 16.* Component-wise latency breakdown of Trifuse (percentage of total inference time, averaged across four benchmarks).

| Method | Model Size | Input Generation (%) | Model Infer (%) | Attention (%) | OCR (%) | Caption (%) | Fusion (%) |
|---|---|---|---|---|---|---|---|
| Trifuse | 3B | 0.83 | 9.69 | 0.74 | 4.84 | 83.89 | < 0.01 |
| Trifuse | 7B | 0.70 | 6.84 | 0.59 | 3.65 | 88.22 | < 0.01 |
| Trifuse+GUI-Actor | 3B | 0.76 | 6.45 | 0.58 | 3.81 | 88.40 | < 0.01 |
| Trifuse+GUI-Actor | 7B | 0.89 | 9.20 | 0.74 | 5.20 | 83.96 | < 0.01 |
| Trifuse+GUI-AIMA | 3B | 0.70 | 6.69 | 0.61 | 4.16 | 87.83 | < 0.01 |

*Table 17.* Ablation studies on two-stage localization for final grounding method.

| Method | ScreenSpot | | | | ScreenSpot-v2 | | | | ScreenSpot-Pro | | OSWorld-G | |
|---|---|---|---|---|---|---|---|---|---|---|---|---|
| | Text | Icon | Avg. | Δ | Text | Icon | Avg. | Δ | Avg. | Δ | Avg. | Δ |
| Direct | 79.8 | 64.1 | 72.7 | - | 81.8 | 67.8 | 75.7 | - | 9.2 | - | 33.1 | - |
| Two-Stage (Trifuse) | 89.9 | 70.5 | 81.1 | **+7.4** | 92.3 | 71.0 | 82.6 | **+6.9** | 18.9 | **+9.7** | 38.7 | **+5.6** |

*Table 18.* Detailed ablation studies on token and head selection strategies across ScreenSpot and ScreenSpot-v2.

*(a) ScreenSpot*

| Method | | Mobile | | Desktop | | Web | | Avg. |
|---|---|---|---|---|---|---|---|---|
| | | Text | Icon | Text | Icon | Text | Icon | |
| | All Head | 18.7 | 7.4 | 17.0 | 12.1 | 3.5 | 1.9 | 10.2 |
| Top Token | Range Head | 23.8 | 14.0 | 23.7 | 18.6 | 7.4 | 1.9 | 14.9 |
| | Top Head | 79.8 | 60.2 | 62.9 | 44.3 | 53.9 | 38.8 | 58.5 |
| | All Token | 61.9 | 39.7 | 49.5 | 39.3 | 39.1 | 19.4 | 42.5 |
| Top Head | Last Token | 73.3 | **62.0** | 59.3 | 37.1 | 49.6 | 29.1 | 53.7 |
| | Top Token | **79.8** | 60.2 | **62.9** | **44.3** | **53.9** | **38.8** | **58.5** |

*(b) ScreenSpot-v2*

| Method | | Mobile | | Desktop | | Web | | Avg. |
|---|---|---|---|---|---|---|---|---|
| | | Text | Icon | Text | Icon | Text | Icon | |
| | All Head | 16.2 | 7.6 | 16.0 | 10.7 | 2.1 | 1.5 | 9.2 |
| Top Token | Range Head | 22.8 | 12.3 | 22.2 | 16.4 | 5.6 | 2.5 | 13.8 |
| | Top Head | 75.2 | 59.7 | 57.2 | 41.4 | 50.9 | 31.5 | 54.7 |
| | All Token | 64.5 | 38.4 | 40.7 | 30.0 | 40.4 | 20.4 | 40.9 |
| Top Head | Last Token | 73.4 | 54.5 | 56.7 | **42.9** | 48.7 | 25.1 | 52.1 |
| | Top Token | 75.2 | 59.7 | 57.2 | 41.4 | **50.9** | **31.5** | **54.7** |

*Table 19.* Detailed ablation studies on fusion strategies across ScreenSpot and ScreenSpot-v2.

*(a) ScreenSpot*

| Method | | Mobile | | Desktop | | Web | | Avg. |
|---|---|---|---|---|---|---|---|---|
| | | Text | Icon | Text | Icon | Text | Icon | |
| Single Modal | OCR | 53.1 | 5.2 | 19.1 | 2.1 | 30.9 | 6.8 | 22.1 |
| | Caption | 33.7 | 32.7 | 17.0 | 11.4 | 36.1 | 19.9 | 26.7 |
| | Attention | 79.8 | 60.2 | 62.9 | 44.3 | 53.9 | 38.8 | 58.5 |
| Fusion | Average Fusion | 88.6 | 47.4 | 72.2 | 40.7 | 71.8 | 40.9 | 63.3 |
| | Custom Fusion | 76.9 | 61.1 | 58.2 | 45.0 | 51.3 | 44.3 | 58.0 |
| | CS Fusion (Trifuse) | **91.9** | **73.4** | **90.2** | **65.7** | **87.4** | **70.4** | **81.1** |

*(b) ScreenSpot-v2*

| Method | | Mobile | | Desktop | | Web | | Avg. |
|---|---|---|---|---|---|---|---|---|
| | | Text | Icon | Text | Icon | Text | Icon | |
| Single Modal | OCR | 47.6 | 3.3 | 15.4 | 2.9 | 27.3 | 3.4 | 19.7 |
| | Caption | 35.9 | 21.8 | 16.5 | 11.4 | 31.3 | 13.1 | 23.2 |
| | Attention | 75.2 | 59.7 | 57.2 | 41.4 | 50.9 | 31.5 | 54.7 |
| Fusion | Average Fusion | 90.8 | 45.8 | 81.4 | 37.9 | 76.1 | 47.6 | 65.8 |
| | Custom Fusion | 90.8 | 45.4 | 73.2 | 35.7 | 69.1 | 42.7 | 62.2 |
| | CS Fusion (Trifuse) | **92.7** | **75.1** | **92.8** | **64.3** | **91.3** | **70.9** | **82.6** |

*Table 20.* Detailed ablation studies on ScreenSpot and ScreenSpot-v2, analyzing the effect of the final grounding strategy across versions.

*(a) ScreenSpot*

| Method | Mobile | | Desktop | | Web | | Avg. |
|---|---|---|---|---|---|---|---|
| | Text | Icon | Text | Icon | Text | Icon | |
| Direct Location | 80.6 | 70.7 | 84.0 | 60.7 | 75.2 | 59.2 | 72.7 |
| Two-Stage (Trifuse) | **91.9** | **73.4** | **90.2** | **65.7** | **87.4** | **70.4** | **81.1** |

*(b) ScreenSpot-v2*

| Method | Mobile | | Desktop | | Web | | Avg. |
|---|---|---|---|---|---|---|---|
| | Text | Icon | Text | Icon | Text | Icon | |
| Direct Location | 82.4 | 71.6 | 82.5 | 65.7 | 80.4 | 65.0 | 75.5 |
| Two-Stage (Trifuse) | **92.7** | **75.1** | **92.8** | **64.3** | **91.3** | **70.9** | **82.6** |

