# OpenReview forum: "Trifuse: Enhancing Attention-Based GUI Grounding via Multimodal Fusion"
_ICML.cc/2026/Conference — ICML 2026 regular_

### Official Review · Reviewer_Vt43 · 2026-03-06

**Soundness:** 3
**Presentation:** 3
**Significance:** 3
**Originality:** 3
**Overall Recommendation:** 4
**Confidence:** 3

**Summary:**

The paper addresses the challenge of Graphical User Interface (GUI) grounding, which involves mapping natural language instructions to specific interface elements. The authors identify that existing training-based methods are data-intensive and generalize poorly to unseen interfaces, while current attention-based (training-free) methods lack reliability due to a deficiency in explicit spatial anchors.

To bridge this gap, the authors propose Trifuse, a training-free, attention-based grounding framework that integrates three complementary modalities：

1. MLLM Attention: Derived through a two-level (token and head) filtering scheme to isolate spatially informative signals.
2. OCR Textual Cues: Extracted text instances aligned semantically with user instructions.
3. Icon-level Caption Semantics: Captures non-textual visual semantics using a pretrained icon detection and captioning model.

These modalities are fused using a Consensus-SinglePeak (CS) strategy designed to emphasize cross-modal agreement while preserving highly discriminative modality-specific peaks. Extensive evaluations on four grounding benchmarks (ScreenSpot, ScreenSpot-v2, ScreenSpot-Pro, and OSWorld-G) demonstrate that Trifuse significantly outperforms existing training-free methods and even approaches the performance of some supervised fine-tuned models.

**Compliance With Llm Reviewing Policy:**

Affirmed.

**Final Justification:**

The authors' rebuttal fully resolved my concerns, reinforcing my prior assessment: 4 weak accept

**Key Questions For Authors:**

1. Given the smaller gains on OSWorld-G due to the "Refusal" category, do you envision a simple way to integrate an infeasibility detector into the fused heatmap (e.g., based on peak confidence scores)?

2. You mentioned the framework is sensitive to its many hyperparameters. Have you tested how well the fixed hyperparameters (e.g., $\tau_{v}=0.5$, $\lambda=0.5$) generalize to completely different interface styles not covered in the four benchmarks?

3. Trifuse involves MLLM inference, OCR, icon detection, captioning, and two-stage zoom-in. What is the typical end-to-end latency compared to a single-pass training-based model, and is this viable for real-time GUI agent interaction?

**Limitations:**

Yes. The authors adequately discuss limitations regarding dependency on auxiliary models, hyperparameter sensitivity, the performance ceiling of training-free approaches, and the lack of a refusal mechanism

**Strengths And Weaknesses:**

**Strengths**

* **Effective Training-Free Paradigm**: Trifuse achieves state-of-the-art performance in the training-free category by leveraging internal MLLM signals combined with off-the-shelf auxiliary tools, reducing reliance on expensive annotated GUI data.

* **Principled Modality Fusion**: The Consensus-SinglePeak (CS) strategy is a key innovation that robustly handles cases where modalities agree (reinforcing consensus) or disagree (trusting the most confident specific peak), avoiding the pitfalls of simple averaging.

* **Robust Attention Extraction**: The two-level filtering strategy (token-level via visual relevance and head-level via spatial entropy) effectively reduces noise in MLLM attention maps, which the authors show is heterogeneous and often non-informative。

**Weaknesses**

* **Dependency on Auxiliary Model Quality**: The performance is inherently capped by the quality of the OCR and icon detection/captioning models. The authors admit that existing icon caption models currently constrain Trifuse's full potential.

* **Coarse Patch Granularity Limitations**: While the two-stage localization helps, the initial grounding is still limited by the coarse spatial granularity of MLLM patch tokens, which can struggle with very small or densely packed elements on high-resolution screens

---

> ### Author Rebuttal · Authors · 2026-03-30
>
> We sincerely thank the reviewer for the constructive comments, which are helpful to improve the quality of our paper. We address each concern directly below and hope these responses adequately resolve the raised concerns.
>
> ---
>
> **Q1: Integrating an Infeasibility Detector into the Fused Heatmap**
>
> Motivated by the reviewer's suggestion, we explored a confidence-based infeasibility detection mechanism: when no region in the fused heatmap exceeds a predefined peak confidence threshold (0.2), the model abstains and outputs a refusal decision. Trifuse* denotes Trifuse equipped with a confidence-based infeasibility detector.
>
> **Table 1. Effect of confidence-based infeasibility detection on OSWorld-G.**
>
> |Method|Model Size|Text Matching(↑)|Element Recognition(↑)|Layout Understanding(↑)|Fine-grained Manipulation(↑)|Refusal(↑)|Average(↑)|
> |:---:|:---:|:---:|:---:|:---:|:---:|:---:|:---:|
> |Trifuse|3B|55.3|33.0|45.6|24.7|0.0|38.7|
> |Trifuse*|3B|54.9|31.2|46.0|24.0|14.8|38.8|
>
> Results on OSWorld-G demonstrate that **this simple method increases Refusal accuracy from 0.0% to 14.8% while keeping overall performance essentially unchanged (38.7% vs. 38.8%)**, suggesting that fused heatmap confidence may carry meaningful signal for abstention in addition to localization. We will include this experiment and discuss infeasibility detection as a promising future direction in the revised paper.
>
> ---
>
> **Q2: Hyperparameter Generalization to Unseen Interface Styles**
>
> We evaluated Trifuse on **UI-Vision** [1], a desktop-centric high-resolution benchmark differing substantially from our original four benchmarks in interface style, using fixed default hyperparameters without any retuning.
>
> **Table 1. Performance on UI-Vision. Higher is better for all metrics. EG denotes element grounding accuracy on three subtasks, while LG reports layout grounding quality (IoU / Precision / Recall). LG evaluates a model's ability to cluster UI elements into functional and semantic groups and predict bounding boxes that encapsulate them.**
>
> |Method|EG-Basic(↑)|EG-Functional(↑)|EG-Spatial(↑)|LG-IoU(↑)|LG-Prec.(↑)|LG-Rec.(↑)|
> |:---:|:---:|:---:|:---:|:---:|:---:|:---:|
> |Qwen2.5-VL-3B|1.0|0.1|0.5|5.0|52.9|5.8|
> |Qwen2.5-VL-7B|1.2|0.8|0.5|5.6|62.1|6.4|
> |TAG-8.5B|10.3|8.0|1.7|1.5|42.8|1.6|
> |Trifuse-3B|12.4|9.7|4.8|5.8|58.1|6.8|
> |Trifuse-7B|**13.6**|**10.3**|**5.8**|**6.0**|**62.6**|**7.1**|
>
> Trifuse outperforms the Qwen2.5-VL backbone and surpasses TAG across all subtasks without retuning. **Additional sweeps over $\tau_v$ and $\lambda$ show that performance variation remains within 0.5% on element grounding tasks,** indicating stable and transferable defaults rather than being narrowly tuned to a particular benchmark. Our conclusion is one of transferability rather than universal invariance, and we will make this scope explicit in the revised paper.
>
> [1] Nayak et al. UI-Vision: A Desktop-centric GUI Benchmark for Visual Perception and Interaction. ICML, 2025.
>
> ---
>
> **Q3: End-to-End Latency**
>
> We conducted a dedicated efficiency evaluation across four benchmarks, randomly sampling 20 examples per benchmark and repeating each measurement 10 times.
>
> **Table 1. Per-image inference time (seconds, mean ± std) and accuracy (%) across four benchmarks.**
>
> |Method|Size|ScreenSpot~Time (↓)|ScreenSpot~Acc. (%, ↑)|ScreenSpot-v2~Time (↓)|ScreenSpot-v2~Acc. (%, ↑)|ScreenSpot-Pro~Time (↓)|ScreenSpot-Pro~Acc. (%, ↑)|OSWorld-G~Time (↓)|OSWorld-G~Acc. (%, ↑)|
> |:---:|:---:|:---:|:---:|:---:|:---:|:---:|:---:|:---:|:---:|
> |TAG|8.5B|4.00 ± 1.04|57.5|4.11 ± 0.76|51.2|4.55 ± 1.84|3.0|6.21 ± 3.99|25.3|
> |Trifuse|3B|7.50 ± 0.67|81.1|7.78 ± 0.70|82.6|8.94 ± 0.94|18.9|8.40 ± 1.15|38.7|
> |Trifuse|7B|7.79 ± 0.99|**86.2**|7.86 ± 0.80|**86.9**|9.99 ± 1.39|**29.7**|8.13 ± 0.96|**47.6**|
>
> **Table 2. Component-wise latency breakdown of Trifuse, reported as percentage of total inference time.**
>
> |Method|Size|Input Generation (%)|Model Infer (%)|Attention (%)|OCR (%)|Caption (%)|Fusion (%)|
> |:---:|:---:|:---:|:---:|:---:|:---:|:---:|:---:|
> |Trifuse|3B|0.83|9.69|0.74|4.84|83.89|<0.01|
> |Trifuse|7B|0.70|6.84|0.59|3.65|88.22|<0.01|
>
> Trifuse incurs approximately **7–10 seconds** of end-to-end latency. **Although Trifuse incurs moderately higher latency than TAG, it achieves substantial accuracy gains**, suggesting a meaningful accuracy–latency trade-off for GUI grounding. The **icon caption model alone accounts for over 83% of total inference time**, while CS fusion contributes negligible overhead.
>
> We acknowledge that the current latency may limit applicability in real-time GUI agent interaction. However, in step-wise GUI agent scenarios where immediate response is not required, a latency of several seconds may remain acceptable. Replacing the caption model with a more lightweight alternative is a promising direction for efficiency improvement, and we will incorporate this full analysis and discussion into the revised paper.

---

> > ### Author Rebuttal · Reviewer_Vt43 · 2026-04-01
> >
> > Thanks for the authors detailed rebuttal, my concerns have been fully resolved.

---

> > > ### Author Response · Authors · 2026-04-02
> > >
> > > Thank you for maintaining your positive score and for your continued support throughout the review process.
> > >
> > > We are delighted that the new experiments effectively addressed your concerns and further strengthened our empirical case. As promised, all new results and discussions will be carefully incorporated into the revised version.
> > >
> > > Thank you again for your invaluable feedback, which has greatly contributed to improving our work!

---

### Official Review · Reviewer_BEgk · 2026-03-08

**Soundness:** 3
**Presentation:** 3
**Significance:** 3
**Originality:** 3
**Overall Recommendation:** 4
**Confidence:** 3

**Summary:**

1. The paper presents Trifuse, a training-free GUI grounding method that fuses MLLM attention, OCR text cues, and icon-caption semantics to map instructions to UI elements.

2. The approach builds modality heatmaps (incl. token/head-filtered attention) and combines them with a Consensus–SinglePeak fusion plus a two-stage zoom-in localization for finer clicks.

3. Experiments on four benchmarks report substantial gains over prior training-free attention baselines and show Trifuse can also act as a plug-in to improve trained models.

**Compliance With Llm Reviewing Policy:**

Affirmed.

**Key Questions For Authors:**

No questions from my side.

**Limitations:**

yes

**Strengths And Weaknesses:**

# Strengths

1. The pipeline is well specified (token/head filtering → three heatmaps → CS fusion → two-stage zoom-in), with explicit equations and clear module separation.

2. Results show large gains over the training-free TAG baseline on four benchmarks, and the same fusion also yields consistent plug-in improvements on trained backbones.

3. The key idea is to explicitly fuse attention + OCR + icon-caption cues via Consensus–SinglePeak, separating cross-modal agreement from modality-specific peaks.

# Weaknesses

1. The qualitative figures in the appendix mostly demonstrate successful cases, but the method undoubtedly has brittle points (e.g., OCR misses, wrong icon captions, attention peaks on distractors). Adding failure cases in the main text would improve the paper’s scientific value by helping readers understand the approach’s limitations and applicable scenarios.

2. Add efficiency numbers. Even basic wall-clock latency per image and a component-wise breakdown would strengthen the paper. Right now the method saves training cost but may impose notable inference cost; the paper should be honest about that trade-off.

---

> ### Author Rebuttal · Authors · 2026-03-30
>
> We sincerely thank the reviewer for the constructive comments, particularly for highlighting the importance of presenting failure cases and efficiency numbers. We address each concern directly below and hope these responses adequately resolve the raised concerns.
>
> ---
>
> **W1: Missing Failure Cases**
>
> We fully agree that presenting failure cases is important for scientific transparency. Through our experiments, **we identified three representative failure patterns**:
>
> - **Caption model failure.** When the icon caption model fails to detect or correctly describe the target icon, it provides no useful fusion signal, which can substantially degrade grounding performance. This is consistent with the limitation discussed in the paper regarding the quality of existing GUI icon caption models.
> - **Distractor interference.** In dense or visually cluttered interfaces, one or more modalities may be attracted to distractor elements, causing fusion to converge on an incorrect region.
> - **Fine-grained localization failure.** In challenging benchmarks such as ScreenSpot-Pro and OSWorld-G, some UI elements are extremely small or visually ambiguous, and the two-stage zoom-in strategy may be insufficient for precise localization in such cases.
>
> We will add representative failure examples illustrating each pattern in the revised paper, accompanied by structured analysis that clarifies Trifuse's boundary conditions. We believe this addition will provide a more balanced and rigorous presentation of the framework's capabilities and limitations.
>
> ------
>
> **W2: Missing Efficiency Numbers**
>
> We conducted a dedicated efficiency evaluation across four benchmarks, randomly sampling 20 examples per benchmark and repeating each measurement 10 times.
>
> **Table 1. Per-image inference time (seconds, mean ± std) and accuracy (%) across four benchmarks.**
>
> | Method | Size | ScreenSpot~Time (↓) | ScreenSpot~Acc. (%, ↑) | ScreenSpot-v2~Time (↓) | ScreenSpot-v2~Acc. (%, ↑) | ScreenSpot-Pro~Time (↓) | ScreenSpot-Pro~Acc. (%, ↑) | OSWorld-G~Time (↓) | OSWorld-G~Acc. (%, ↑) |
> | :---: | :---: | :---: | :---: | :---: | :---: | :---: | :---: | :---: | :---: |
> | TAG | 8.5B | 4.00 ± 1.04 | 57.5 | 4.11 ± 0.76 | 51.2 | 4.55 ± 1.84 | 3.0 | 6.21 ± 3.99 | 25.3 |
> | Trifuse | 3B | 7.50 ± 0.67 | 81.1 | 7.78 ± 0.70 | 82.6 | 8.94 ± 0.94 | 18.9 | 8.40 ± 1.15 | 38.7 |
> | Trifuse | 7B | 7.79 ± 0.99 | **86.2** | 7.86 ± 0.80 | **86.9** | 9.99 ± 1.39 | **29.7** | 8.13 ± 0.96 | **47.6** |
>
> **Table 2. Component-wise latency breakdown of Trifuse, reported as percentage of total inference time.**
>
> | Method | Size | Input Generation (%) | Model Infer (%) | Attention (%) | OCR (%) | Caption (%) | Fusion (%) |
> | :---: | :---: | :---: | :---: | :---: | :---: | :---: | :---: |
> | Trifuse | 3B | 0.83 | 9.69 | 0.74 | 4.84 | 83.89 | <0.01 |
> | Trifuse | 7B | 0.70 | 6.84 | 0.59 | 3.65 | 88.22 | <0.01 |
>
> **Although Trifuse incurs moderately higher latency than TAG, it achieves substantial accuracy gains**, suggesting a meaningful accuracy–latency trade-off for GUI grounding.
>
> The breakdown reveals that the **icon caption model accounts for over 83% of total inference time**, consistent with the limitation discussed in the paper. By contrast, the CS fusion strategy introduces negligible overhead (<0.01%), suggesting it is not a runtime bottleneck. We will incorporate this full efficiency analysis and an explicit discussion into the revised paper.

---

### Official Review · Reviewer_5GiY · 2026-03-11

**Soundness:** 4
**Presentation:** 4
**Significance:** 3
**Originality:** 3
**Overall Recommendation:** 5
**Confidence:** 4

**Summary:**

The authors propose **Trifuse**, an attention-based, predominantly training-free framework for Graphical User Interface (GUI) grounding. The paper addresses a known limitation in current Multimodal Large Language Models (MLLMs): while internal attention mechanisms contain some localization signals, they lack the explicit, complementary spatial anchors necessary for reliable zero-shot GUI grounding.

**Compliance With Llm Reviewing Policy:**

Affirmed.

**Final Justification:**

The authors have conducted two more important experiments on efficiency analysis and hyperparameters serach, demonstrate the trade-off between latency and accuracy, and the trade-off is acceptable according to the gains. All my concerns have be resolved, therefore, I have raised my score to accept, I think this paper is a good contribution to the GUI Agent community.

**Key Questions For Authors:**

**Questions:**

1. See weakness 1, extra Efficiency experiments to show the privilege of Trifuse as a training-free method.

2. See weakness 2, hyperparameters studies.


**Minor Suggestions / Typos:**
There is a large blank gap under Table 7 (lines 395-400). Removing this empty space will improve the overall layout and visual presentation of the paper.

**Limitations:**

yes

**Strengths And Weaknesses:**

**Strengths**:

* 1. The CS fusion mechanism is elegantly designed. By explicitly calculating a consensus term via element-wise multiplication alongside a dynamically weighted single-peak term, the framework successfully mitigates the noise typical of simple averaging ensembles.
* 2. The paper provides a valuable diagnostic observation that not all attention heads in MLLMs are spatially informative. The use of spatial entropy to filter diffuse attention heads is a rigorous, mathematically sound approach to signal extraction without fine-tuning.
* 3. **Strong Empirical Performance:** For a framework requiring no task-specific fine-tuning, the results are highly competitive. It substantially outperforms the standard training-free baseline, TAG, across ScreenSpot, ScreenSpot-v2, and ScreenSpot-Pro. Furthermore, the framework acts as a modular enhancement, improving the accuracy of existing training-based models like GUI-Actor and GUI-AIMA when integrated.


**Weakness**:

* **1. Lack of analysis on Efficiency:** The two-stage zoom-in localization strategy requires running inference on the full image, cropping, resizing, and running inference a second time. Coupled with the need to execute three distinct vision models to extract the modalities, the latency and computational cost at inference time are likely substantial, the paper could be enhanced if the authors can conduct an evaluation on the efficiency analysis to show whether Trifuse also have advantages on efficiency compared with other training-free methods?

* **2. As the authors also mentioned, they admit that the framework is highly sensitive to a large number of hyperparameters.** Can you provide the grid-search hyperparameters experiment to show the consistency of Trifuse? (e,g. strict top-k cutoffs, threshold quantiles etc..), that will confirm that the model may not be brittle and heavily overfitted to the validation distributions of the tested benchmarks.

---

> ### Author Rebuttal · Authors · 2026-03-30
>
> We sincerely thank the reviewer for the thorough and careful reading of our manuscript. The questions on efficiency and hyperparameter sensitivity pushed us to conduct more rigorous analysis. We address each concern directly below, providing latency measurements, component-wise breakdowns, and sensitivity results. We hope these responses adequately address the raised concerns.
>
> ---
>
> **Q1: Lack of Efficiency Analysis**
>
> We sincerely thank the reviewer for this important suggestion. We conducted a dedicated efficiency evaluation across four benchmarks, randomly sampling 20 examples per benchmark and repeating each measurement 10 times.
>
> **Table 1. Per-image inference time (seconds, mean ± std) and accuracy (%) across four benchmarks.**
>
> | Method | Size | ScreenSpot~Time (↓) | ScreenSpot~Acc. (%, ↑) | ScreenSpot-v2~Time (↓) | ScreenSpot-v2~Acc. (%, ↑) | ScreenSpot-Pro~Time (↓) | ScreenSpot-Pro~Acc. (%, ↑) | OSWorld-G~Time (↓) | OSWorld-G~Acc. (%, ↑) |
> | :---: | :---: | :---: | :---: | :---: | :---: | :---: | :---: | :---: | :---: |
> | TAG | 8.5B | 4.00 ± 1.04 | 57.5 | 4.11 ± 0.76 | 51.2 | 4.55 ± 1.84 | 3.0 | 6.21 ± 3.99 | 25.3 |
> | Trifuse | 3B | 7.50 ± 0.67 | 81.1 | 7.78 ± 0.70 | 82.6 | 8.94 ± 0.94 | 18.9 | 8.40 ± 1.15 | 38.7 |
> | Trifuse | 7B | 7.79 ± 0.99 | **86.2** | 7.86 ± 0.80 | **86.9** | 9.99 ± 1.39 | **29.7** | 8.13 ± 0.96 | **47.6** |
>
> **Although Trifuse incurs moderately higher latency than TAG, it achieves substantial accuracy gains**, suggesting a meaningful accuracy–latency trade-off for GUI grounding. To identify the bottleneck, we performed a component-wise breakdown averaged across four benchmarks:
>
> **Table 2. Component-wise latency breakdown of Trifuse, reported as percentage of total inference time.**
>
> | Method | Size | Input Generation (%) | Model Infer (%) | Attention (%) | OCR (%) | Caption (%) | Fusion (%) |
> | :---: | :---: | :---: | :---: | :---: | :---: | :---: | :---: |
> | Trifuse | 3B | 0.83 | 9.69 | 0.74 | 4.84 | 83.89 | <0.01 |
> | Trifuse | 7B | 0.70 | 6.84 | 0.59 | 3.65 | 88.22 | <0.01 |
>
> The **icon caption model accounts for over 83% of total inference time** across all configurations, consistent with the limitation discussed in the paper. Regarding GPU memory, the OCR module adds ~200 MB on average and the caption model ~40 MB on average, which is modest relative to the 3B/7B backbone.
>
> We view developing a more efficient GUI icon caption model as the most impactful direction for reducing runtime, and will incorporate this full analysis and discussion into the revised paper.
>
> **Q2: Hyperparameter Sensitivity**
>
> We conducted additional sensitivity analysis over the main hyperparameters on ScreenSpot. The complete analysis across all benchmarks will be incorporated into the revised paper.
>
> **Table 1. Sensitivity to $\tau_s$ and $\lambda$ on ScreenSpot (accuracy, %).**
>
> | TAG (↑) | Hyperparameter |              Values              |          Trifuse-3B (↑)          |          Trifuse-7B (↑)          |
> | :-----: | :------------: | :------------------------------: | :------------------------------: | :------------------------------: |
> |  57.5   |    $\tau_s$    | 0.70 / 0.75 / 0.80 / 0.85 / 0.90 | 79.8 / 80.2 / 80.8 / 80.3 / 79.6 | 84.7 / 85.3 / 85.7 / 85.2 / 84.7 |
> |  57.5   |   $\lambda$    |   0.3 / 0.4 / 0.5 / 0.6 / 0.7    | 80.2 / 80.9 / 81.1 / 81.0 / 80.8 | 85.3 / 85.5 / 86.2 / 86.1 / 85.7 |
>
> Performance varies smoothly across all ranges without sharp discontinuities: ±2% for $\tau_s$, ±1.5% for $\lambda$, and ±1.2% for $\alpha$–$\beta$. **Even under the least favorable configurations, Trifuse remains substantially above TAG (57.5%)**. These results suggest moderate variation rather than brittleness. Due to space constraints, full tables are omitted here but will be incorporated into the revised paper alongside threshold quantile experiments for each modality. We agree that “sensitivity” may overstate the effect, and we will revise the manuscript description appropriately.
>
> ---
>
> **Q3: Minor Suggestions / Typos**
>
> We sincerely thank the reviewer for the comprehensive and careful review of the manuscript. The blank gap below Table 7 is unintentional and will be corrected in the revised version.
>
> ---
>
> **References**
>
> [1] Kang et al. Your large vision-language model only needs a few attention heads for visual grounding. CVPR, 2025.
>
> [2] Zhou et al. GUI-AIMA: Aligning Intrinsic Multimodal Attention with a Context Anchor for GUI Grounding. arXiv, 2025.

---

> > ### Author Rebuttal · Reviewer_5GiY · 2026-04-02
> >
> > Thanks for the authors extra efficiency and hyperparameter search experiments. As authors also notice, although Trifuse incurs moderately higher latency than TAG, the accuracy gains are quite promising. Hope the authors can add these experiments to the revised manuscript.
> >
> > All my concerns have be resolved, therefore, I have raised my score to accept, I think this paper is a good contribution to the GUI Agent community.

---

> > > ### Author Response · Authors · 2026-04-02
> > >
> > > Thank you for your thorough evaluation and for raising your score to accept. We are glad that the efficiency and hyperparameter experiments addressed your concerns. As promised, additional experiments will be incorporated into the revised manuscript.
> > >
> > > Thank you again for your kind recognition of our work's contribution to the GUI Agent community. Your insightful feedback has been invaluable in helping us strengthen the paper!

---

### Official Review · Reviewer_Lcrf · 2026-03-11

**Soundness:** 3
**Presentation:** 3
**Significance:** 3
**Originality:** 2
**Overall Recommendation:** 5
**Confidence:** 4

**Summary:**

This paper proposes Trifuse, a training-free GUI grounding framework that fuses three complementary modalities, i.e., MLLM attention, OCR-derived text cues, and icon caption semantics, through a Consensus-SinglePeak (CS) fusion strategy. The method applies token- and head-level filtering for attention extraction, and a two-stage zoom-in localization. Evaluated on four benchmarks, Trifuse substantially outperforms prior training-free methods and provides gains when combined with training-based approaches.

**Compliance With Llm Reviewing Policy:**

Affirmed.

**Final Justification:**

The authors have addressed my concerns so I have raised my score to 5.

**Key Questions For Authors:**

Please refer to the weaknesses mentioned above. The reviewer is willing to raise the score if they are addressed satisfactorily.

**Limitations:**

Yes

**Strengths And Weaknesses:**

# Strengths
S1: **Strong empirical results in the training-free setting.** The improvement over TAG is very large (e.g., +23.6 on ScreenSpot, +31.4 on ScreenSpot-v2), and Trifuse with a 3B model approaches or exceeds many supervised 7B models. This is a meaningful practical contribution, as it reduces reliance on expensive GUI-annotated data.

S2: **Well-designed ablation studies.** The paper systematically ablates each component: token filtering, head filtering, individual modalities, and fusion strategies. Table 6 and Table 7 clearly show the value of each design choice. The hyperparameter sensitivity analysis in Figure 3 and the two-stage localization ablation (Table 8) are also helpful.

S3: **Generality as a modular enhancement.** Trifuse can be layered on top of training-based attention methods (GUI-Actor, GUI-AIMA), consistently improving accuracy. This demonstrates its value as a complementary module rather than a standalone competitor.

# Weaknesses

W1: **Limited novelty in individual components.** Each component of Trifuse is individually well-established. The OCR and caption pipelines are straightforward embedding similarity computations. The two-stage localization is explicitly acknowledged as borrowed from prior work. The primary novelty is the CS fusion strategy, but the consensus term is simply element-wise multiplication, and the single-peak term is a confidence-weighted aggregation with a sigmoid gate. The technical depth needs to be questioned somehow.

W2: **Heavy reliance on off-the-shelf tools raises questions about the "training-free" framing.** While Trifuse itself requires no GUI-specific fine-tuning, it depends on PaddleOCR, OmniParser (which was trained on GUI data), and BGE-M3. OmniParser, in particular, was designed for GUI understanding and was trained on GUI-related data. The paper's framing as "training-free" is somewhat misleading, as it's more accurately "no additional GUI grounding fine-tuning," since the auxiliary models bring substantial domain knowledge. This is acknowledged implicitly (the paper notes OmniParser achieves 73.0 on ScreenSpot by itself in Table 2), but the distinction deserves more explicit discussion.

W3: **Missing evaluation on desktop high-resolution benchmarks.** The paper claims Trifuse is effective across diverse platforms and resolutions. While the benchmarks include desktop categories, they lack a dedicated desktop high-resolution benchmark that specifically isolates dense, high-native-resolution desktop environments. A desktop-focused high-resolution benchmark such as UIVision[1] would better validate the claimed generalization, particularly since desktop environments feature denser layouts, smaller UI elements, and higher native resolutions that stress both the attention extraction and the two-stage localization components. ScreenSpot-Pro partially covers professional software but mixes platforms, while dedicated desktop high-resolution benchmarks would more rigorously test whether the CS fusion and zoom-in strategy truly scales to the fine-grained localization demands of real-world desktop usage. Including such a benchmark would strengthen the paper's claims about resolution robustness and cross-platform generality.

[1] Nayak et al. UI-Vision: A Desktop-centric GUI Benchmark for Visual Perception and Interaction

W4: **A large number of hyperparameters and acknowledged sensitivity.** The authors themselves note in Appendix D that the framework is sensitive to hyperparameters. While the fixed settings work across benchmarks, this raises concerns about how well these transfer to new GUI domains not covered by the evaluation benchmarks. No cross-validation or robustness analysis to hyperparameter perturbations is provided.

---

> ### Author Rebuttal · Authors · 2026-03-30
>
> We sincerely thank the reviewer for the constructive comments, which are helpful to improve the quality of our paper. We address each concern directly in our responses below, providing clarifications, additional analysis, and supplementary results. We hope these responses adequately resolve the concerns raised.
>
> ---
>
> **W1: Limited Novelty in Individual Components**
>
> The primary contribution of our paper is to **build a simple yet effective multimodal fusion framework for GUI grounding, which has not been explored in existing research.**
>
> Prior attention-based methods predominantly rely on a single modality [1], leaving complementary signals from OCR and caption cues unexploited. **Trifuse** integrates all three modalities within a unified framework, and experimental results clearly demonstrate that CS fusion consistently outperforms attention-only baselines across all benchmarks. We believe more advanced modality designs, fusion strategies, and localization methods are promising directions for future work.
>
> ---
>
> **W2: Misleading "Training-Free" Framing**
>
> Following prior research [2], we used "training-free" to denote the absence of task-specific fine-tuning for the grounding model, not the absence of pre-trained components. We agree this framing deserves explicit clarification. In the revised paper, **we will replace "training-free" with "without task-specific fine-tuning"** and provide clear discussion of what this claim does and does not cover. We will also add complete OmniParser results across four benchmarks for better comparison.
>
> ---
>
> **W3: Missing Desktop High-Resolution Benchmark**
>
> Following the reviewer's recommendation, we evaluated Trifuse on UI-Vision, a dedicated high-resolution desktop benchmark with dense layouts and small UI elements.
>
> **Table 1. Performance on UI-Vision. EG denotes element grounding accuracy on three subtasks, while LG reports layout grounding quality (IoU / Precision / Recall). LG evaluates a model's ability to cluster UI elements into functional and semantic groups and predict bounding boxes that encapsulate them.**
>
> | Method | EG-Basic (↑) | EG-Functional (↑) | EG-Spatial (↑) | LG-IoU (↑) | LG-Prec. (↑) | LG-Rec. (↑) |
> | :---: | :---: | :---: | :---: | :---: | :---: | :---: |
> | Qwen2.5-VL-3B | 1.0 | 0.1 | 0.5 | 5.0 | 52.9 | 5.8 |
> | Qwen2.5-VL-7B | 1.2 | 0.8 | 0.5 | 5.6 | 62.1 | 6.4 |
> | TAG-8.5B | 10.3 | 8.0 | 1.7 | 1.5 | 42.8 | 1.6 |
> | Trifuse-3B | 12.4 | 9.7 | 4.8 | 5.8 | 58.1 | 6.8 |
> | Trifuse-7B | **13.6** | **10.3** | **5.8** | **6.0** | **62.6** | **7.1** |
>
> On UI-Vision, **Trifuse outperforms the Qwen2.5-VL backbone and surpasses TAG on all subtasks**, suggesting that CS fusion and zoom-in localization remain competitive in high-resolution desktop settings.
>
> ---
>
> **W4: Hyperparameter Sensitivity and Robustness**
>
> We conducted additional perturbation analysis covering major hyperparameters on ScreenSpot, and additionally evaluated $\tau_v$ and $\lambda$ on UI-Vision to assess cross-domain transferability. Due to space constraints, we present representative results here and will include the complete analysis in the revised paper.
>
> **Table 1. Sensitivity to $\tau_s$ and $\lambda$ on ScreenSpot (accuracy, %).**
>
> | TAG (↑) | Hyperparameter |              Values              |          Trifuse-3B (↑)          |          Trifuse-7B (↑)          |
> | :-----: | :------------: | :------------------------------: | :------------------------------: | :------------------------------: |
> |  57.5   |    $\tau_s$    | 0.70 / 0.75 / 0.80 / 0.85 / 0.90 | 79.8 / 80.2 / 80.8 / 80.3 / 79.6 | 84.7 / 85.3 / 85.7 / 85.2 / 84.7 |
> |  57.5   |   $\lambda$    |   0.3 / 0.4 / 0.5 / 0.6 / 0.7    | 80.2 / 80.9 / 81.1 / 81.0 / 80.8 | 85.3 / 85.5 / 86.2 / 86.1 / 85.7 |
>
> Performance varies smoothly across all ranges: approximately ±2% for $\tau_s$, ±1.5% for $\lambda$, and ±1.2% for the $\alpha$–$\beta$ combination. **Notably, even under the least favorable configurations, Trifuse remains substantially above TAG (57.5%).** On UI-Vision, variation across all tested values of $\tau_v$ and $\lambda$ remains within 0.5% on element grounding tasks, suggesting that the default settings transfer reasonably well. We agree that “sensitivity” may overstate the effect, and we will revise the manuscript description appropriately.
>
> ---
>
> **References**
>
> [1] Wu et al. GUI-Actor: Coordinate-Free Visual Grounding for GUI Agents. NeurIPS, 2025.
>
> [2] Xu et al. Attention-driven GUI grounding: Leveraging Pretrained Multimodal Large Language Models without Fine-Tuning. AAAI, 2025.

---

> > ### Author Rebuttal · Reviewer_Lcrf · 2026-04-03
> >
> > Thank the authors for the detailed reply, which has basically addressed my concerns. So I have raised my score to 5. Please include the content in the rebuttal into the later version to make it more rigorous.

---

> > > ### Author Response · Authors · 2026-04-04
> > >
> > > Thank you for your positive feedback and for raising the score to 5. We are pleased that our rebuttal addressed your concerns. As promised, all the content discussed in the rebuttal will be incorporated into the revised manuscript to further strengthen its rigor.
> > >
> > > Thank you again for your constructive feedback!

---

### Decision · Program_Chairs · 2026-04-30

**Decision:**

Accept (regular)

**Comment:**

This paper proposed the Trifuse approach for the GUI grounding problem. The Trifuse is a training-free approach that leveraging visual attention, OCR-derived textual cues, and icon-level caption semantics for improving the grounding quality.

All the reviewers agree to accept this paper with quite solid scores: 2x Strong Accept and 2x Weak Accept.

I will vote for acceptance.